# Modelling feedbacks between the Northern-Hemisphere ice sheets and climate during the last glacial cycle

Meike D. W. Scherrenberg[1], Constantijn J. Berends[1], Lennert B. Stap[1], Roderik S.W. van de Wal[1,2]

[1]Institute for Marine and Atmospheric research Utrecht, Utrecht University, 3584 CC Utrecht, the Netherlands
[2]Faculty of Geosciences, Department of Physical Geography, Utrecht University, Utrecht, the Netherlands

*Correspondence to*: M.D.W. Scherrenberg (M.D.W.Scherrenberg@uu.nl)

**Abstract.** During the last glacial cycle (LGC), ice sheets covered large parts of Eurasia and North America which resulted in ~120 meters of sea level change. Ice sheet – climate interactions have considerable influence on temperature and precipitation patterns, and therefore need to be included when simulating this time period. Ideally, ice sheet – climate
interactions are simulated by a high-resolution earth system model. While these models are capable of simulating climates at a certain point in time, such as the pre-industrial (PI) or the last glacial maximum (LGM; 21,000 years ago), a full transient glacial cycle is currently computationally unfeasible as it requires a too large amount of computation time. Nevertheless, ice-sheet models require forcing that captures the gradual change in climate over time to calculate the accumulation and melt of ice and its effect on ice sheet extent and volume changes.

Here we simulate the LGC using an ice sheet model forced by LGM and PI climates. The gradual change in climate is modelled by transiently interpolating between pre-calculated results from a climate model for the LGM and the PI. To assess the influence of ice sheet – climate interactions, we use two different interpolation methods: The climate matrix method, which includes a temperature-albedo and precipitation-topography feedback, and the glacial index method, which does not. To investigate the sensitivity of the results to the prescribed climate forcing, we use the output
of several models that are part of the Paleoclimate Modelling Intercomparison Project Phase III (PMIP3). In these simulations, ice volume is prescribed and the climate is reconstructed with a general circulation model (GCM). Here we test those models by using their climate to drive an ice sheet model over the LGC.

We find that the ice volume differences caused by the climate forcing exceed the differences caused by the interpolation method. Some GCMs produced unrealistic LGM volumes and only four resulted in reasonable ice sheets
with LGM Northern Hemisphere sea-level contribution ranging between 74 – 113 meters with respect to the present day. The glacial index and climate matrix methods result in similar ice volumes at LGM, but yield a different ice evolution with different ice domes during the inception phase of the glacial cycle, and different sea-level rates during the deglaciation phase. The temperature-albedo feedback is the main cause of differences between the glacial index and climate matrix methods.

## 1 Introduction

Sea-level rise due to the melt of the Greenland and Antarctic ice sheets is one of the biggest threats posed by anthropogenic climate change (Fox-Kemper et al., 2021). Ice sheets have a substantial influence on the climate system, as these can amplify changes in temperatures and alter precipitation patterns, which in turn affect ice accumulation and ablation. It is therefore important to make accurate projections of future sea level change, that include the interactions between the climate and the ice sheets. These interactions are important on long time scales because ice sheets respond generally slowly to changes in temperature and precipitation. (Clark et al., 1999), although brief periods of rapid ice loss have occasionally occurred in the geological past (e.g., Gomez et al., 2015; Brendryen et al., 2020). Direct observations are insufficient to study ice sheet – climate interaction as these only capture the changes in ice sheets over the past century. Instead, the paleo-record provides information on the climate system, such as ice sheet extent and thickness, atmospheric $CO_2$ concentrations and eustatic sea level, dating back well before modern observations. These paleo-reconstructions allow the study of considerable changes in the ice sheet and climate that took place over multiple millennia. The most recent period in the Earth's history with substantial changes in ice sheet extent is the Last Glacial Cycle (LGC; 120,000 – 8,000 years ago). This period is associated with a sea level change of approximately 120 meters (e.g., Simms et al., 2019, Gowan et al., 2021) and a decrease in global temperature of 4 – 6 K (e.g., Annan et al., 2022, Tierney et al., 2020) with respect to present day.

The climate during the LGC is affected by several internal and external processes in the climate system. Over long time scales, insolation changes due to orbital parameters are significant enough to affect climate and ice sheets (Löfverström et al., 2014). Additionally, atmospheric $CO_2$ concentrations changed between 190 and 280 ppm during the LGC, also acting as a forcing to the climate system. Topography and albedo changes result from the change in ice extent and thickness affect temperature and precipitation (e.g., Abe-Ouchi et al., 2007, Clark et al., 1999). Especially the temperature-albedo and precipitation-topography interactions induced by changes in ice volume have a substantial impact on ice sheets (Abe-Ouchi et al., 2007, Stap et al. 2014). For example, a decrease in temperature prompts an increase in snow coverage and thereby albedo. This increase in albedo decreases the total thermal energy stored in the climate system, as a larger portion of solar radiation entering the atmosphere is reflected towards space. Therefore, the change in albedo causes a positive feedback enhancing temperature change on regional and global spatial scales. In addition, local temperatures are also affected by changes in surface topography due to the atmospheric temperature lapse rate.

In addition, topography has a local and regional impact on precipitation. Precipitation is enhanced on the slopes of ice margins, since cooling and condensation takes place when air is lifted from the margin to the inland ice. During this transport, air cools and moisture is removed resulting in low precipitation on the lee side and on ice plateaus. Changes in surface topography during glacial cycles can also affect atmospheric circulation (e.g, Löfverström et al., 2016), again

affecting temperature and precipitation patterns (Pausata et al., 2011, Ullman et al., 2014). These feedbacks, which act over multiple millennia as the ice sheet gradually incepts, grows, and retreats, must be accounted for during transient climate and ice sheet simulations.

An ideal set-up to transiently simulate ice-sheet climate interactions during the LGC would involve a fully coupled general circulation model (GCM) that simulates ice sheets, oceans and atmosphere. However, these simulations require a large amount of computation time, making them currently unfeasible. One strategy to deal with this excessive computational demand is to decrease grid resolution, use asynchronous coupling while certain physical processes are artificially accelerated (e.g., Smith & Gregory, 2012) or use earth system models of intermediate complexity which have

a reduced computational cost compared to GCM's, but can still explicitly simulate ice-sheet climate interactions (Ganopolski et al., 2010). Alternatively, when simulating specific periods in time such as the last glacial maximum (LGM; 21,000 years ago) and pre-industrial (PI), GCMs can be used. Modelling efforts such as the Paleoclimate Modelling Intercomparison Project (PMIP) intercompare a set of GCMs that used similar boundary conditions, such as atmospheric $CO_2$ concentration, orbital configuration, and fixed prescribed ice-sheet geometry, to simulate the climate

for a specific time slice. The third phase of PMIP, PMIP3 (Braconnot et al., 2011), used nine GCMs to simulate climates during LGM and PI. Each climate model used prescribed ice sheets by Abe Ouchi et al. (2015), hence transient changes in ice topography were not simulated. Despite using the same boundary conditions, ice-sheet model studies using the PMIP3 models such as Niu et al., (2019) and Alder & Hostetler, (2019) found substantial differences in LGM ice volume and extent. In these studies, Niu simulated the entire last glacial cycle while Adler & Hostetler used steady state LGM

climate forcing, both finding large differences between the ice sheets resulting from the GCM climates, some showing unrealistic ice sheets.

       Simulating a full glacial cycle using a stand-alone ice-sheet model is feasible as these models require only a small amount of run-time compared to GCMs. GCMs model the entire globe and simulate processes that require small time steps. Ice-sheet models can run at a much lower temporal resolution and only require a grid that covers a small

portion of the Earth, but they need a higher spatial resolution.

       In the past different methods have been used to include transiently changing climate forcing to an ice-sheet model. These include applying a globally uniform temperature anomaly directly to present-day climate (De Boer, 2014), by interpolating between LGM, mid-Holocene and PI surface mass balances from a climate model (Fyke et al., 2014), by coupling the ice-sheet to a simple energy balance model (Stap et al., 2014), using earth system models of intermediate

complexity (Ganopolski et al., 2010), or by using a glacial index method (Niu et al., 2019). In the glacial index method LGM and PI temperature and precipitation are transiently interpolated with respect to one parameter obtained from paleo-reconstructions, such as $CO_2$. Therefore, as $CO_2$ decreases, the climate cools and dries as it approaches the LGM. This method has been shown to yield LGM ice volume and extent that agrees well with reconstructions (e.g, Charbit et

al., 2007, Niu et al., 2019). However, this interpolation does not include any ice sheet – climate interactions, other than (usually) a lapse-rate based linear elevation correction. To parameterize these feedbacks, we can use a climate matrix method instead (Pollard., 2010). For this we need results from at least two time slices, although more will provide better constraints. Hence with a climate matrix method the ice sheet evolution effects the dynamics of the climate and ice system whereas with a glacial index the ice sheet only responds passively to the climate forcing. The climate matrix method implicitly resolves the temperature-albedo and precipitation-topography feedbacks. Here, the climate time slices are interpolated with respect to both prescribed forcing, for example atmospheric $CO_2$ concentrations, and the internally calculated albedo and ice sheet geometry. Since it incorporates calculated fields from the ice sheet model into the interpolation, a climate matrix method can implicitly resolve ice sheet – climate feedbacks requiring only a small amount of additional computational resources. When performing realistic scenarios of the LGC, the climate matrix method was shown to be able to successfully replicate ice evolution (Berends et al., 2018).

While the climate matrix method and glacial index method have been used in the past, a comparison with a realistic scenario for the LGC has not yet been explored. Ladant et al. (2014) simulated ice sheets during the Eocene-Oligocene transition and Abe-Ouchi et al. (2013) investigated Pleistocene glacial cycles by interpolating between snapshots with varying ice sheet sizes, orbital parameters and atmospheric $CO_2$. Recently, Stap et al. (2022) compared a matrix and index method with schematic experiments of Antarctica during the Miocene. This showed that the temperature-albedo and precipitation-topography feedback operate in opposite ways: The temperature-albedo feedback substantially reduces, and precipitation-topography feedback slightly increases glacial-interglacial variability. They suggested that the temperature-albedo feedback is stronger than the precipitation-topography feedback.

Here we aim to build upon the work by Stap et al. (2022) by intercomparing a glacial index and climate matrix method using a realistic experiment by simulating the Northern Hemisphere ice sheets during LGC. We force the ice-sheet model using different available climate model output from the PMIP3 ensemble. This study uses a climate matrix method, therefore also builds on Niu et al. (2019), who simulated the LGC using PMIP3 climate forcing interpolated with a glacial index method. Our ice-sheet model and climate forcing set-up are described in section 2. Section 3 introduces our simulations of the LGC and shows the differences due to climate forcing and due to the interpolation methods. These findings are discussed in section 4.

## 2. Methods

### 2.1 Ice sheet model

In this study, we use the three-dimensional thermodynamically-coupled ice-sheet model IMAU-ICE version 2.0 (Berends et al., 2022). This model uses the depth-integrated viscosity approximation (DIVA; Goldberg, 2011) to

calculate the dynamics of floating and grounded ice. This vertically-integrated approximation to the stress balance is similar to the hybrid shallow ice / shallow shelf approximation (Bueler and Brown, 2009), but has improved physics, is more efficient and has improved numerical stability (Robinson et al., 2022). A regularised Coulomb sliding law is used to calculate basal friction (Bueler & van Pelt, 2015). Proper grounding-line migration is achieved using a sub-grid friction-scaling scheme, which is based on the approach used in the Parallel Ice Sheet Model (PISM; Feldmann et al., 2014) and the Community Ice Sheet Model (CISM; Leguy et al., 2021). Sub-shelf melt rates are calculated using a temperature and depth-dependent sub-shelf melt parameterization (Martin et al. 2011) in combination with parameterised, globally uniform ocean temperature changes (De Boer et al., 2013). As a result, while computationally fast, this parameterization does not include ocean temperature fields from the GCM simulations, and also does not capture the spatial pattern in ocean temperature changes. Bedrock adjustment to changes in ice load is modelled using an Elastic Lithosphere, Relaxing Asthenosphere model (Le Meur and Huybrechts, 1996). Calving is not included in this version of the model.

We simulate the North American, Eurasian and Greenland ice sheets concurrently in three separate domains with a 40x40km, 40x40km, and 20x20km resolution respectively (see Fig. 1). We use a higher resolution for Greenland as the ice sheet is smaller, allowing us to capture the effect of small topographical changes with a comparative number of grid-cells compared to North America and Eurasia. To prevent double counting of ice, ice growth is prevented in the Greenlandic parts of the North American and Eurasian domains and Ellesmere Island in the Greenland domain. Antarctica is not included in these simulations as the feedbacks from topography and albedo throughout the LGC are small compared to Eurasia and North America, due to the relatively small change in extent and elevation. In reality, Antarctica may have a substantial impact on ocean circulation and the carbon cycle (Adkins, 2013) and thereby climate evolution, but these feedbacks can only be captured in fully coupled models and not with the parameterized forcing applied here.

## 2.2 Surface mass balance

The monthly surface mass balance (SMB) is calculated from the monthly temperature and precipitation resulting from the climate interpolation (see 2.4) using an insolation-temperature model (IMAU-ITM; Berends et al., 2018), which has been part of the Greenland Surface Mass Balance Model Intercomparison Project (GrSMBMIP; Fettweis et al., 2020). In this parameterized SMB scheme, snow accumulation is calculated using a precipitation and temperature-dependent snow-rain partitioning by Ohmura, (1999). This snow-rain partitioning was tuned towards the regional climate model RACMO as part of GrSMBMIP. Annual refreezing is calculated using the approach by Huybrechts and de Wolde, (1999) and Janssens and Huybrechts, (2000). Ablation is parameterised and depends on temperature, insolation and surface

albedo (Bintanja et al., 2002). The insolation at the top of the atmosphere is prescribed and obtained from Laskar et al. (2004). Surface albedo is calculated internally: First a snow-free albedo is applied, with 0.1 for ocean, 0.2 for land and 0.5 for bare ice. A firn layer is added on top which, depending on depth, can increase the albedo to a maximum of 0.85. The planetary albedo changes due to cloud cover changes is not taken into account. The equations governing IMAU-ITM are presented in Appendix A.

## 2.3 PMIP3 climate time slices

We obtained climate forcing for near-surface air temperature, precipitation and topography from the nine available LGM and PI GCM simulations that are part of PMIP3. While recently several LGM and PI simulations for PMIP4 have become available, we nevertheless decided to use PMIP3. PMIP4 is not fundamentally different to PMIP3 (Kageyama et al., 2021). However, PMIP3 has been more widely studied, allowing for comparisons with studies that have been conducted in the past such as Niu et al. (2019) and Alder & Hostetler (2019). The PMIP3 simulations are listed in Table 1. Each of these models used identical boundary conditions following the PMIP3 protocol, such as orbital parameters, trace gases, and ice sheets. The PMIP3 protocol contains a prescribed ice sheet by Abe-Ouchi et al., 2015 which is a composite of the ICE-6G v2.0 (Argus & Peltier, 2010), GLAC-1a (Tarasov et al., 2012) and ANU (Lambeck et al., 2010). We selected a subset of the nine available PMIP3 LGM and PI simulations to obtain climate forcing that result in good agreement with reconstructions. This step is described in Appendix B. The selection of acceptable models consists of COSMOS, IPSL, MIROC and MPI. Our parameterized SMB scheme was tuned to the mean climate of this sub-selection, which is hereafter referred to as PMIP3-Ensemble.

To apply the GCM fields to the ice sheet model, corrections need to be applied to account for the difference in resolution and topography. First a correction is applied to deal with biases in the GCM models, which is described in Appendix C. Secondly, to account for the large difference in resolution, the GCM fields are bilinearly interpolated to the ice-sheet model grid. Thirdly, since the ice-sheet model topography evolves during the LGC, which is not accounted for in the GCM climate time-slices, a topographic correction needs to be applied. Here we use the approach by Berends et al. (2018), which implies that temperature is downscaled using a dynamic lapse rate correction. Precipitation is downscaled using the Roe and Lindzen (2001) model. This model accounts for the precipitation changes as a result of large-scale changes in topography, and takes in account the surface slope, wind direction and changes in surface height. Throughout the LGC, Greenland's topography change is less substantial compared to North America and Eurasia. Therefore, instead of the Roe and Lindzen model, we apply a Clausius-Clapeyron that adapts the applied precipitation based on the change in surface temperature.

## 2.4 Transiently changing forcing

In this study we use two methods to transiently interpolate between climate time slices: a glacial index method and a climate matrix method. In our glacial index method, precipitation and temperature fields are interpolated with respect to prescribed $CO_2$ obtained from Bereiter et al. (2015). Fig. 2 depicts the atmospheric $CO_2$ concentration during the LGC and the corresponding values for the glacial index. A glacial index of 1 (0) represents full glacial (interglacial) conditions. Hence, the climate forcing is equal to LGM (PI). Temperature and precipitation are respectively interpolated linearly

and logarithmically. The logarithmical interpolation for precipitation prevents negative values and is used to describe the relative changes during the LGC.

The climate matrix method expands upon the glacial index method by implicitly resolving the temperature-albedo and precipitation-topography feedbacks. This is achieved by making the interpolation parameter spatially variable. In this study, we applied the climate matrix method using the approach by Berends et al. (2018). Temperature

is interpolated with respect to $CO_2$, and to the absorbed insolation at the surface. The absorbed insolation is computed using the internally calculated albedo, and the insolation solution by Laskar et al. (2004). Therefore, any modelled advance of the ice will result in an increased albedo, an increase in the interpolation weight, and therefore a more glacial climate. Precipitation is interpolated with respect to the change in topography between LGM and PI. Regions without LGM ice cover do not undergo substantial topographical changes. In these regions precipitation is interpolated with

respect to the total changes of topography throughout the domain. Equations governing the glacial index and climate matrix methods are presented in Appendix D. Neither the climate matrix nor the glacial index method includes interactions between the ocean and ice or climate. However, changes in the ocean circulation had substantial influence on the climate, the carbon cycle (Adkins, 2013; Toucanne et al., 2021), and affected sub-shelf melt rates. Atmospheric circulation changes due to ice sheet topography changes are also not explicitly modelled. Nevertheless, the climate matrix

method is a computationally fast method to implicitly resolve ice-temperature and topography-precipitation feedbacks without transient GCM simulations. This method should therefore be seen as an alternative to the glacial index method, not an alternative to climate models.

## 3. Results

### 3.1 Climate forcing

In this section, we compare simulations of the LGC resulting from different climate forcings using the climate matrix method. Fig. 3 shows the sea-level contribution during the LGC with the climate forcing from COSMOS, IPSL, MIROC, MPI, and the PMIP3-Ensemble. The sea level contribution at LGM ranges substantially between the five simulations.

As shown in Fig. 3a, the LGM sea level contribution of the Northern Hemisphere ice sheets ranges between 74 (COSMOS) and 119 (MIROC) meters of sea-level equivalent (m.s.l.e.). When including the contribution for Antarctica
of 10 m.s.l.e. (Simms et al., 2019), the total LGM global sea level contribution is within range of Simms et al. (2019) for the PMIP3-Ensemble, MPI and IPSL.

The North American ice sheet is the largest contributor to the differences in ice volume with a minimum LGM contribution of 56 m.s.l.e. (COSMOS) and a maximum of 87 m.s.l.e. (MIROC; Fig. 3b). For the Eurasian ice sheet, LGM sea level contribution varies between 12 m (IPSL) and 29 m (MIROC) (Fig. 3c). This substantial difference
between ice sheet model runs forced by individual PMIP3-members is in line with findings by Niu et al., (2019) and Alder & Hostetler, (2019).

Fig. 4 shows when each region was covered by ice for the first time during the LGC, thereby indicating the timing of inception and the gradual increase in extent of the ice sheets. For example, dark purple areas over Ellesmere Island and Baffin Island show the immediate inception of the ice sheets in those regions. They are currently covered by
ice caps and therefore near pre-industrial temperatures are sufficient for ice growth. The light orange colours in southern Scandinavia indicate that ice covered these regions from ~30 ka onwards. In North America, the ice sheets incept from the North American Cordillera and Eastern Canada forming two large ice domes. In Eurasia, ice sheets incept at the islands surrounding the Barents Sea. Little evidence exists on the inception phase of the LGC, since later glaciations tend to have removed most geological evidence. Modelling studies based on geological evidence, such as Bahadory et
al. (2021) and Dalton et al. (2022), suggest that the North American ice sheet incepted at Ellesmere Island and the Canadian Cordillera, which agrees reasonably well for most GCM forcings except for COSMOS forcing.

Between 120 and 60 ka, simulations forced with MPI and IPSL output have a comparatively large sea level contribution (Fig. 3a). This can also be seen in the ice evolution (Fig. 4) for MPI and IPSL forcing leading to a comparatively large extent of the Baffin and Innuitian ice sheets during the inception phase of the ice sheets. Similarly,
MPI forcing resulted in a large extent of the Barents-Kara ice sheets at the onset of the LGC. Since each simulation was forced by the same prescribed $CO_2$ and insolation, this is a result of the climate forcing and internal feedback processes.

Fig. 5 shows the LGM-PI temperature differences in the ice-sheet model domains. Shown here is that regions with large ice extent at the early parts of the simulations correspond generally well to large LGM-PI temperature differences promoting ice growth. The LGM and PI temperatures are linearly interpolated with respect to prescribed $CO_2$
and albedo. Therefore, the same change in $CO_2$ and albedo results in a larger temperature change with increasing LGM-PI temperature difference provided by the GCMs.

Fig. 6 depicts ice thickness at LGM, showing that the ice extent varies considerably with climate forcing. In North America, the differences in extent are mostly located along the southern margins. As the ice sheet forced with MIROC has an unrealistic large extent compared to reconstructions, the LGM ice sheet exceeds the southern domain

boundary. In the Eurasian ice sheet, the differences are found in Western Europe. Although in reality the British Isles were covered by glacial ice at LGM (e.g., Abe-Ouchi et al., 2015, Batchelor et al., 2019), this only occurs in the simulation forced by the MIROC climate. Though this is accompanied with a large ice volume compared to the reconstructions by Simms et al., 2019 (see Fig. 3d). IPSL and COSMOS forcing lead to limited LGM ice extent south of the Scandinavian mountains, which does not match paleo-reconstructions (e.g., Abe-Ouchi et al., 2015, Batchelor et

al., 2019). Fig. 7 shows LGM temperature and compares it to the extent of the ice sheet simulations. This figure indicates that low LGM summer temperatures tend to match ice extent well, which is in line with Niu et al. (2019). This is to be expected considering that temperature is important for the SMB by affecting the amount of melt, refreezing and snowfall.

We are able to capture the LGM sea level (Simms et al., 2019) and extent (Abe-Ouchi et al., 2015) well using the climate

matrix method. While we tuned ice volume towards Simms et al. (2019), the extent of the LGM ice sheet is smaller compared to Abe-Ouchi et al. (2015). However, the rate of ice growth during Marine Isotope Stage (MIS)5, as well as peak ice volumes at 60 ka are not captured well (e.g., Gowan et al., 2021). This may be a result of a too small effect of insolation on temperature and SMB. The interior of the Laurentide ice sheet in our simulations has a thickness over 4000 meters, exceeding the thickness of the ICE6G-C reconstruction by Peltier et al. (2015). This suggests that not enough

ice is transported from the interior towards the margin, resulting in a small extent and large thickness in the interior, possibly a result of shortcomings in the basal sliding and specifically a too high basal friction.

**3.2 Ice sheet – climate feedbacks**

Here we investigate the effect of the albedo-temperature and topography-precipitation feedbacks. We present two simulations using the PMIP3-Ensemble forcing, which differ in their use of either the glacial index or the climate matrix

method.

Fig. 8c and 8d compare when the first ice accumulates in regions for the glacial index and climate matrix methods. In the Eurasian ice sheet, which incepts mostly in the Barents-Kara Sea region, more domes are formed in the glacial index method. This difference in the number of domes is even more pronounced in North America. Using the climate matrix

method, only the Laurentide and Cordilleran ice domes develop, which merge around 40 ka. In the glacial index method, many smaller domes are formed in the Keewatin, Baffin and Cordilleran regions. The ice domes in the glacial index method simulations have much more irregular shapes compared to the smoother margins of the climate matrix ice domes. This North American ice sheet with few inception regions in the climate matrix method agrees better with studies conducted with geological constraints (e.g., Batchelor et al., 2019, Dalton et al., 2022).

275       This difference in inception between the glacial index method and the climate matrix method is due to the feedback processes. An ice sheet has a high albedo, and thereby creates a regionally cold climate enhancing ice growth. In regions without snow or ice, albedo is low leading to higher temperatures inhibiting ice inception. This feedback process is included in the climate matrix method, which separates temperature change caused by albedo and insolation, and $CO_2$. Consequently, the albedo in the ice-sheet model has a pronounced influence on temperature and thereby on the

ice-sheet evolution. In the glacial index method temperature is only affected by $CO_2$ and does not account for albedo changes. Hence, the glacial index method underestimates cooling in high albedo regions and overestimates cooling in low albedo regions.

       Fig. 9 shows the sea level contribution of the Northern Hemisphere ice sheets over time. Ice volume at LGM is slightly larger in the glacial index method (110 m.s.l.e.) compared to the climate matrix method (96 m.s.l.e.).

Accordingly, the glacial index method has larger volumes for both the North American (Fig. 9b) and Eurasian (Fig. 9c) ice sheets. Ice thickness at LGM is shown in Fig. 8a,b. The Eurasian ice sheet in the glacial index method has more ice covering the British Islands, but less in the Barents Sea region. The North American ice sheet margin extents more towards the south in the glacial index method. The smaller volume in the climate matrix method is mostly caused by the temperature-albedo interaction. In the climate matrix method, temperature and precipitation are only equal to the LGM

time slice when $CO_2$, albedo and topography are the same as the ice sheet reconstruction. Since the ice sheet extent in the climate matrix method is smaller than LGM compared to the reconstruction by Abe-Ouchi et al. (2015). Albedo is too low in the region and consequentially the temperature is too high. Therefore, at LGM, the temperature in the ice-sheet model is higher compared to the LGM climate forcing. In the glacial index method, when $CO_2$ levels are equal to LGM the climate forcing is equal to LGM as well. Therefore, in these simulations, the LGM volume is higher in the

glacial index method.

       After the LGM, the Eurasian and North American ice sheets retreat. Fig. 10b shows the modelled sea level contribution rate during the LGC. While sea-level contribution rates reach negative values at approximately the same time, the climate matrix method finishes retreating later (5 ka) compared to the glacial index method (8 ka). Furthermore, Fig. 10a compares sea-level contribution to $CO_2$, with timestamps indicated in the figure. Considering an arbitrary

threshold of 5 mm/yr shows that the glacial index method retreat rate accelerates faster at lower $CO_2$ concentrations (18 ka, 225 ppm) compared to the climate matrix method (16 ka, 240 ppm). This indicates a lower $CO_2$ threshold for retreat in the glacial index method. The peak sea-level rate is higher and earlier in the glacial index method, with a decrease of 32 mm/yr (11 ka) compared to 19 mm/yr (8 ka) in the climate matrix method. This is because when the $CO_2$ increases rapidly, low temperatures persist longer in the climate matrix method due to the high albedo of the large ice sheet, while

the response of the glacial index method is instantaneous. The climate matrix has the tendency to maintain the ice sheet as it is, and does not warm as quickly, resulting in lower retreat rates.

## 4. Discussion and Conclusion

In this study we simulated the Northern Hemisphere ice sheets of the LGC using an ice-sheet model. Our aim was to investigate the sensitivity to paleoclimate forcing, and to assess the effects of the albedo-temperature and precipitation-topography interactions.

Climate forcing is obtained from the PMIP3 ensemble to investigate the sensitivity of a climate matrix and glacial index method. We find that the differences in ice volume due to the climate forcing exceeds the differences caused by the transient climate interpolation method (glacial index versus climate matrix). Several PMIP3 models yield unrealistic ice sheet configurations. Despite choosing only a subset of the PMIP3 simulations, the sea-level contribution of the Northern-Hemisphere ice sheets still shows a considerable range of 74 – 113 meters. These large differences are in line with findings by Niu et al. (2019) and Alder & Hostetler (2019), and are not exclusively found with PMIP3 forcing, but also for the first PMIP ensemble by Charbit et al. (2007). Our study additionally used the climate matrix method instead of only the glacial index method used by Charbit et al. (2007) and Niu et al. (2019), showing that even when including atmospheric feedback processes, the range in sea level contribution is still large. As originally suggested by Niu et al. (2019), cold LGM summer temperatures generally correspond well with areas of LGM ice cover when using PMIP3 climate forcing to simulate the LGC. This is caused by the fact that high ablation rates at the margin inhibit ice advance and ablation rates depends strongly on temperature.

In this study, we compared the glacial index and climate matrix method with a realistic experiment for the first time. The LGC simulations using the glacial index and climate matrix method were conducted with the same climate forcing, namely the PMIP3 ensemble mean. The glacial index method and climate matrix methods yield similar LGM ice volumes, but show a much larger difference in ice evolution. Generally, the climate matrix method incepts at fewer domes, with the few domes that form being larger and more regularly shaped compared to the glacial index method. In the glacial index method, many more smaller domes form, often far away from the main ice sheet centres. In both methods, the domes gradually merge to form one big Eurasian or Laurentide ice sheet. The difference in inception is due to the albedo-temperature feedback, where temperatures in the climate matrix method are low along ice margins and high in regions with low albedo. In the climate matrix method this enhances ice growth close to ice margins and inhibits ice inception of new domes in regions with low albedo. We find that the ice sheets, especially the Laurentide ice sheet, deglaciate faster in the glacial index method compared to the climate matrix method. This is attributed to the albedo-temperature feedback, as the high albedo of the ice sheet can maintain colder temperatures at the onset of deglaciation.

LGM extent matches reconstructions well, but there are discrepancies for the British Islands. This may have been a result of our simplistic approach to sub shelf melt rates. Furthermore, Greenland and North America are simulated in separate domains, preventing ice dynamical interactions between these ice sheets. Additionally, temperatures in the climate forcing may have been too high for ice inception in the British ice sheet, except for the simulation forced by

climate from MIROC. Neither the glacial index nor climate matrix method match global eustatic sea level evolution well during MIS5 compared to reconstructions (e.g., Gowan et al., 2021). In either method ice volume increases gradually, while studies that are based on geological constrains such as Batchelor et al., 2019 and Dalton et al., 2022 suggest a more dynamic change in ice volume and area for the North American and Eurasian ice sheets. Our method is unable to capture the maximum volume and extent of the Eurasian ice sheets at 60 ka (e.g., Gowan et al., 2021; Kleman et al., 2013). This indicates that we are missing forcing and or feedback processes that would improve ice evolution. One of the main reasons could be a too weak effect of insolation on temperature and SMB. In both the climate matrix and glacial index methods $CO_2$ is the main driver of temperature changes. However, the $CO_2$ record alone is not sufficient to capture the variability in ice evolution throughout the LGC. Insolation was also found to have a substantial impact on glacial cycles (Abe-Ouchi et al., 2013).

In addition, the climate matrix method can be expanded by including snapshots that were generated with different ice sheets and orbital parameters. With more snapshots, a large range of orbital parameters and ice sheets can be studied (e.g., Abe-Ouchi et al., 2013, Ladant et al., 2014). Here we have chosen to use a wide arrange of GCM simulations, limiting the availability of different snapshots such as insolation minima and maxima. As a consequence of using only LGM and PI time slices we do not capture threshold behaviour such as the closure of the Canadian Artic Archipelago gateways (Löfverström et al., 2022), abrupt changes in atmospheric circulations in the North Atlantic during deglaciation (Löfverström et al., 2017), or processes that are not captured in LGM and PI time slices such as Dansgaard/Oeschger and Heinrich events and their impact on climate (Claussen et al., 2003). While we included parameterised albedo-temperature and precipitation-topography interactions, there are many more ice-sheet – climate interactions that are not included. Ice sheet can affect atmospheric circulation and vice versa, which is not well resolved in either glacial index or climate matrix methods, especially given the limited number of GCM snapshots. For example, the North American ice sheets affects atmospheric circulations which can alter the location and size of the Eurasian ice sheet (Liakka et al., 2016), with a large North American ice sheet preventing growth of ice in Siberia. Currently, this feedback is not resolved, which may have affected the evolution and shape of the Eurasian ice sheet throughout the LGC. Fresh water influx into the ocean as the Laurentide ice sheet melted may have had a large impact on the Atlantic meridional overturning circulation (Otto-Bliesner, et al., 2010). In our method, the ocean is not explicitly modelled. Our basal melt method is simplistic and parameterized and does not use ocean data from a GCM. Therefore, this method is unable to capture any ice-ocean interactions, nor the effect of ocean circulation changes. In the climate matrix method, albedo is annually averaged, ignoring seasonal fluctuations. To explicitly simulate the above-mentioned processes, transient GCM simulations are required, which perform adequately, rather than any temporal interpolation method as used here. This requires a major investment of computational resources. The climate matrix method should therefore be viewed as an alternative to the glacial index method, not an alternative to climate models.

## Appendix A: Surface mass balance model

We use IMAU-ITM to calculate surface mass balance (Berends et al., 2018). This SMB scheme requires temperature and precipitation fields that we obtain from the GCM fields after applying the methods described in appendix C and D. Monthly melt in IMAU-ITM is parameterised using Bintanja et al. (2002) and calculated as following:

$$Melt(x,y,m) = (c_1\,(T(x,y,m) - T0) + c_2\,(1 - \alpha(x,y,m))\,Q_{TOA}(x,y,m) - c_3)(s/yr)/\,(L_{fusion}\,12000), \quad (1)$$

Here, the melting temperature of ice ($T0$) is 273.16 K, $Q_{TOA}$ is the insolation at the top of the atmosphere from Laskar et al. (2004) in W/m$^2$ and $L_{fusion}$ is the latent heat of fusion in J/kg. Values for $c_1$, $c_2$ and $c_3$ are tuning parameters shown in Table A1. Albedo ($\alpha$) is calculated internally, based on the approach by Bintanja et al., 2002:

$$\alpha_{surface}(x,y,m) = \alpha_{snow} - (\alpha_{snow} - \alpha_{background})e^{-15\,FirnDepth(x,y,m-1)} - 0.015\,MeltPreviousYr(x,y), \quad (2)$$

Here, $\alpha_{background}$ is 0.1 for water, 0.2 for land, 0.5 for bare ice, and $\alpha_{snow}$ is 0.85 and MeltPreviousYr is the melt of the previous model year. The albedo is capped between the background and snow albedo. Firn depth is calculated using the amount of snowfall that has been added without melting and is capped at 10 meters. For snowfall we use a temperature-based snow rain partitioning from Ohmura et al. (1999).

$$snowfraction = 0.5\left(1 - \frac{atan\frac{(T(x,y,m)-T0)}{3.5}}{1.25664}\right), \quad (3)$$

The $snowfraction$ is the fraction of precipitation (P) that falls as snow with the remainder falling as rain. To calculate the total accumulation, we need refreezing. This is calculated using the approach by Huybrechts & de Wolde, 1999:

$$SuperImposedWater(x,y,m) = \max\{0, 0.012(T0 - T(x,y,m))\}, \quad (4)$$

$$LiquidWater(x,y,m) = RainFall(x,y,m) + Melt(x,y,m), \quad (5)$$

$$Refreezing(x,y,m) = \min\{\min\{SuperImposedWater(x,y,m), LiquidWater(x,y,m)\}, P(x,y,m)\}, \quad (6)$$

Using the previous calculated snowfall, rainfall, refreezing and melt, the SMB can be obtained:

$$SMB(x,y,m) = Snowfall(x,y,m) + Refreezing(x,y,m) - Melt(x,y,m). \quad (7)$$

**Table A1.** The parameters used to calculate ablation in the approach by Bintanja et al. (2002). The surface mass balance parameters were tuned towards the PMIP3-Ensemble simulation to obtain ice sheets that agree well with Simms et al., 2019.

| Domain | c1 ($m\,yr^{-1}\,K^{-1}$) | c2 ($m^3 J^{-1}$) | c3 ($m^3 J^{-1}$; Preliminary Experiments) | c3 ($m^3 J^{-1}$; Tuned) |
|---|---|---|---|---|
| North America | 10 | 0.513 | 18 | 18 |
| Eurasia | 10 | 0.513 | 34.5 | 20 |
| Greenland | 10 | 0.513 | 24 | 24 |

**Appendix B: Climate forcing selection and SMB tuning**

Niu et al. (2019) and Alder & Hostetler (2019) have shown that there are large differences in the modelled ice sheets between the different PMIP3 climates. For our ice sheet simulations, we would ideally use climate forcing with which we can obtain a good representation of the ice sheets at LGM. First, we conducted simulations forced with all nine available PMIP3 simulations using an untuned ablation (see Table A1), these are shown in Figure B1. Since we obtained large differences between the simulations forced by the individual PMIP3 ensemble members, some deviating

substantially from reconstructions, we used a smaller selection of models.

To make a selection of the PMIP3 climates, we compared the simulated ice sheet extent to the reconstruction from Abe-Ouchi et al. (2015). This reconstruction was used as a prescribed LGM ice extent in the climate model simulations. We compare the ice sheet model and reconstruction at LGM to compute the percentage of too large and too small ice extent. For example, the simulation with MRI forcing resulted in a small Eurasian ice sheet. Ice in the simulation

does not cover 91% of the reconstructed extent. Some ice is also found outside the extent of the reconstruction. Therefore, MRI climate forcing also resulted in 1% too large ice extent outside the reconstruction. These percentages for too large and too small ice sheets are listed in Table B1. We added these values together and applied a threshold of 40% to select the climate forcing. MIROC, IPSL, COSMOS and MPI stayed below this threshold for both the Eurasian and North American ice sheets. The other models are above this threshold and are excluded from further study. A climate forcing

was made from the mean of this selection and called PMIP3-Ensemble (4 GCMs). Using this ensemble climate, the SMB model was tuned towards sea level by Simms et al. (2019).

**Table B1.** Shown here are the percentage of ice extent that is too large or too small compared to the reconstruction by Abe-Ouchi et al. (2015). These percentages are added to obtain a quality assessment for the preliminary simulations. Lower percentages indicate a better match to Abe-Ouchi et al. (2015). Simulations that stayed below the 40% threshold are depicted in bold.

| Climate Model | Eurasia Too Small | Eurasia Too Large | Eurasia Total | North America Too Small | North America Too Large | North America Total |
|---|---|---|---|---|---|---|
| CCSM4 | 0.32 | 0.21 | 0.53 | 0.89 | 0.00 | 0.89 |
| CNRM-CM5 | 0.97 | 0.00 | 0.97 | 0.99 | 0.00 | 0.98 |
| **COSMOS-ASO** | 0.18 | 0.15 | **0.33** | 0.33 | 0.00 | **0.33** |
| FGOALS-g2 | 0.11 | 0.37 | 0.48 | 0.44 | 0.01 | 0.45 |
| GISS-E2-R | 0.13 | 1.37 | 1.50 | 0.77 | 0.03 | 0.79 |
| **IPSL-CM5A-LR** | 0.24 | 0.04 | **0.28** | 0.26 | 0.00 | **0.26** |
| **MIROC-ESM** | 0.06 | 0.28 | **0.34** | 0.17 | 0.02 | **0.19** |
| **MPI-ESM-P** | 0.10 | 0.16 | **0.26** | 0.25 | 0.00 | **0.25** |
| MRI-CGCM3 | 0.91 | 0.01 | 0.92 | 0.97 | 0.00 | 0.97 |


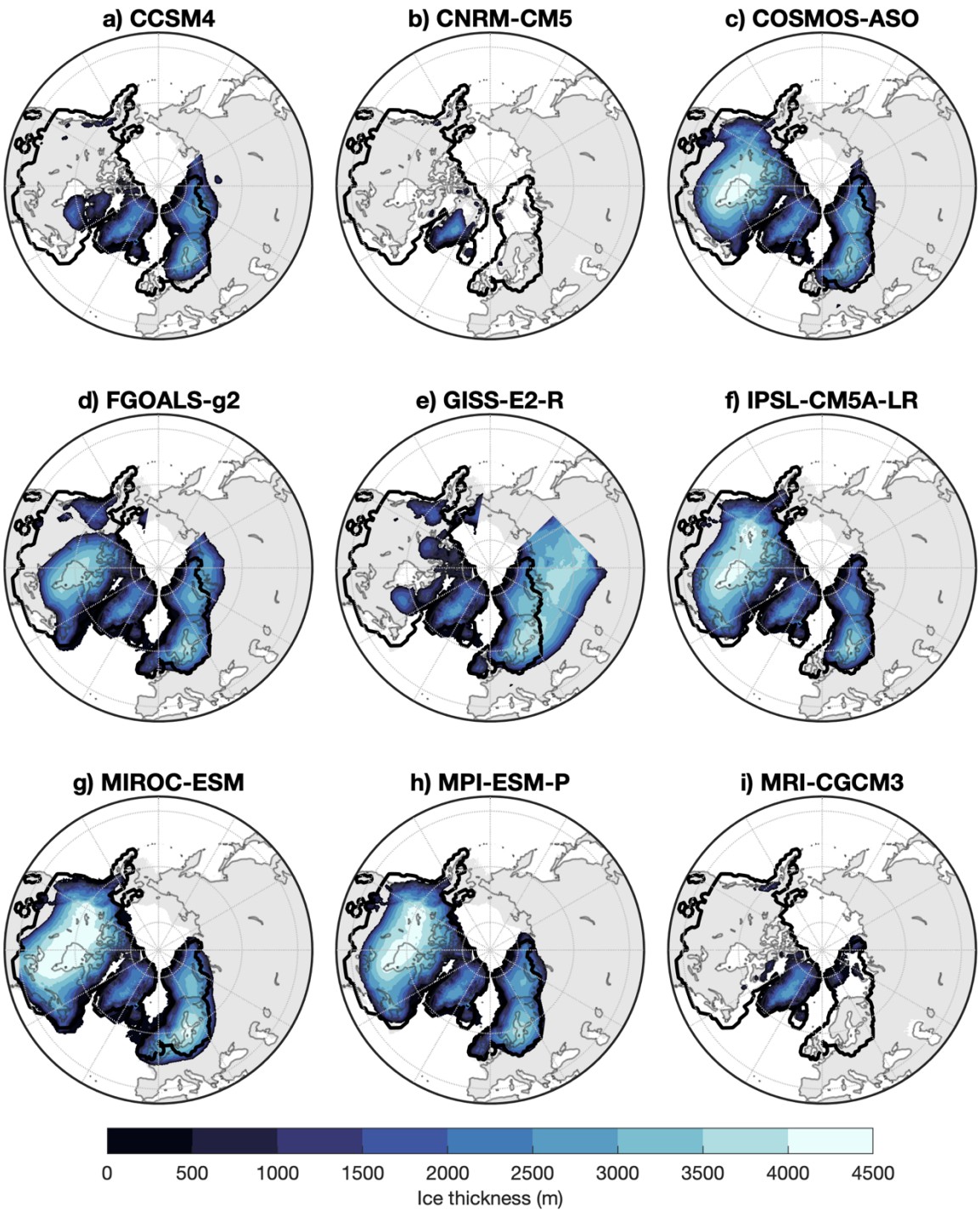

**Figure B1.** LGM ice thickness of an ice sheet model forced by nine different PMIP3 climates, interpolated using the climate matrix method. Black contours represent to the LGM ice sheet reconstruction by Abe Ouchi et al. (2015).

## Appendix C: Bias correction

We apply a correction to the climate forcing to correct for biases in the climate models. First, we calculate the difference between the PI climate from the General Circulation Models (GCM) and observed climate from ERA40 (Uppala et al., 2005). These differences are then applied to the LGM and PI climate forcing.

To apply the bias correction for temperature, we first need to account for differences in topography between the model and observation data. Here we apply a lapse rate ($\lambda$) correction to calculate temperature at sea level.

$$T_{obs,SL}(x,y,m) = T_{obs,PD}(x,y,m) + Hs_{obs,PD}(x,y)\,\lambda(x,y), \tag{8}$$

$$T_{GCM,SL}(x,y,m) = T_{GCM,PD}(x,y,m) + Hs_{GCM,PD}(x,y)\,\lambda(x,y), \tag{9}$$

Using the temperature at sea level from observations ($T_{obs,SL}$) and the climate model ($T_{GCM,SL}$), we can calculate the

difference between observed and modelled temperature.

$$T_{GCM,bias}(x,y,m) = T_{GCM,SL}(x,y,m) - T_{obs,SL}(x,y,m), \tag{10}$$

$T_{GCM,bias}$ represents the bias between modelled and observed data. We can apply this to the LGM and PI time slices to obtain a bias-corrected temperature.

$$T_{GCM,corr}(x,y,m) = T_{GCM}(x,y,m) - T_{GCM,bias}(x,y,m), \tag{11}$$

Here, $T_{GCM}$ is the GCM output, either PI or LGM, while $T_{GCM,corr}$ the bias corrected GCM data.

For precipitation (P), we use the ratio between model and observed data instead. First, we calculate the bias between model and observational data directly.

$$P_{GCM,bias}(x,y,m) = P_{GCM,PI}(x,y,m) \,/\, P_{obs,PI}(x,y,m), \tag{12}$$

Secondly, we apply this bias correction on the GCM data.

$$P_{GCM,corr}(x,y,m) = P_{GCM}(x,y,m) \,/\, P_{GCM,bias}(x,y,m), \tag{13}$$

After applying these corrections, we have obtained bias corrected temperature and precipitation fields for LGM and PI.

## Appendix D:  Climate interpolation

The glacial index and climate matrix methods are used to interpolate between the climate model LGM and PI time slices. As a result, both interpolation methods produce a transiently changing climate throughout the LGC. Here we apply the

method by Berends et al. (2018), for which we use the bias corrected fields for temperature and precipitation. To compute the interpolated temperature and precipitation forcing in both the glacial index and climate matrix method, we use the following two equations:

$$T_{ref}(x,y,m) = w_{tot}(x,y)\,T_{PI,corr}(x,y,m) + \big(1 - w_{tot}(x,y)\big)\,T_{LGM,corr}(x,y,m), \tag{14}$$

$$P_{ref} = e\left(w_{PI}(x,y)\log\left(P_{PI,corr}(x,y,m)\right) + w_{LGM}(x,y)\log\left(P_{LGM,corr}(x,y,m)\right)\right), \tag{15}$$

Here $T_{ref}$ and $P_{ref}$ are the temperature and precipitation climate forcing respectively. The weights for the interpolation are represented by $w_{tot}$, $w_{LGM}$ and $w_{PI}$. The calculation for these interpolation weights determines the difference between the climate matrix and glacial index method.

In the glacial index method $w_{tot}$, $w_{LGM}$ and $w_{PI}$ are calculated with only prescribed $CO_2$.

$$w_{CO2} = (CO_2 - CO2_{LGM}) / (CO2_{PI} - CO2_{LGM}), \tag{16}$$

Here, $CO_2$ represents the prescribed $CO_2$ concentration obtained from Bereiter et al. (2015) at the current model time-step. LGM and PI $CO_2$ concentrations are 190 and 280 ppm respectively. Note here that $w_{CO2}$ is a scalar. In the glacial index method, $w_{CO2}$ is equal to $w_{tot}$, $w_{PI}$, and , $(1 - w_{LGM})$. Since $CO_2$ is the same across the entire globe, not only are these values uniform, but they are also the same for each domain. There is no interaction between the ice sheet model and the interpolation.

The climate matrix method expands on the glacial index method by including a parameterised temperature-albedo and precipitation-topography feedback. For temperature, $w_{tot}$ is calculated using both prescribed $CO_2$ and absorbed insolation. The annual absorbed insolation is calculated as following:

$$I_{abs}(x,y) = \sum_{m=1}^{12} Q_{TOA}(x,y,m)\left(1 - \alpha_{surface}(x,y,m)\right), \tag{17}$$

Here $Q_{TOA}$ is the insolation at the top of the atmosphere, for which we use prescribed forcing from Laskar et al. (2004).

Albedo ($\alpha$) is calculated internally, more details on this is described in appendix A. To calculate annual $I_{abs}$, we use the sum of the monthly $I_{abs}$. We can use this to calculate the interpolation weights for the absorbed insolation.

$$w_{ins}(x,y) = (I_{abs}(x,y) - I_{abs,LGM}(x,y)) / (I_{abs,PI}(x,y) - I_{abs,LGM}(x,y)), \tag{18}$$

$I_{abs_{PI}}$ and $I_{abs_{LGM}}$ are the reference PI and LGM absorbed insolation respectively. This reference albedo is calculated using equation 2, where we use PI and LGM for $Q_{TOA}$ and $\alpha_{surface}$. To calculate albedo at LGM, we apply the

topography, ice mask, and land mask from Abe-Ouchi et al. (2015) and run the SMB model to obtain the albedo change due to snow coverage. Since albedo and insolation vary across the domain, $w_{ins}$ is not uniform. To obtain the regional effect of albedo on temperature, we calculate the mean $w_{ins}$ ($w_{ins,av}$) and $w_{ins}$ with a gaussian smoothening of 200 km ($w_{ins,smooth}$). Using these insolation weights, we can calculate the interpolation weight ($w_{ice}$) resulting from ice interactions.

In the North America and Eurasia domain:

$$w_{ice}(x,y) = \frac{w_{ins}(x,y) + 3\,w_{ins,smooth}(x,y) + 3\,w_{ins,av}(x,y)}{7}, \tag{19}$$

$$w_{tot,T}(x,y) = \frac{(w_{CO2}(x,y) + w_{ice}(x,y))}{2}, \tag{20}$$

In the Greenland domain, we employed a slightly different method. In Eurasia and North America, the change in albedo is mainly due a change in snow coverage and the increase in ice extent. The Greenland ice sheet is covered by ice during both PI and LGM. Albedo changes due to the expansion of ice shelves. Therefore, we interpolate with respect to the total change in albedo for the Greenland domain instead.

$$w_{ice}(x,y) = \frac{w_{ins,smooth}\,(x,y) + 6\,w_{ins,av}}{7}, \tag{21}$$

$$w_{tot,T}(x,y) = \frac{(3\,w_{CO2}(x,y)+w_{ice}(x,y))}{4}, \tag{22}$$

The field $w_{tot}$ is used in equation 14 to calculate temperature.

Precipitation in the climate matrix method is only dependent on surface topography. First, we calculate the total change in surface topography ($Hs$) between LGM and PI.

$$w_{tot,P} = \left(\sum Hs\,(x,y) - \sum Hs_{PI}(x,y)\right) / \left(\sum Hs_{LGM}(x,y) - \sum Hs_{PI}(x,y)\right), \tag{23}$$

For regions within the ice extent of Abe-Ouchi et al. (2015), we calculate the regional change in topography.

$$w_{LGM,Hs}(x,y) = \frac{Hs_{ISM}(x,y) - Hs_{GCM\,PI}(x,y)}{Hs_{GCM,LGM}(x,y) - Hs_{GCM,PI}(x,y)}\,w_{tot,P}(x,y), \tag{24}$$

In regions that are outside the bounds of this ice extent $w_{LGM,Hs}$ is equal to $w_{tot,P}$. We multiply $w_{LGM}$ with $w_{tot}$ to account for the regional effect on precipitation due to a change in surface topography.

$$w_{LGM}(x,y) = w_{LGM,Hs}(x,y)\,w_{tot,P}(x,y), \tag{25}$$

This $w_{LGM}$ is used in equation 15 to calculate precipitation. Note here that precipitation is not uniform. Only as the ice sheet increases in thickness, Hs increases which raises $w_{LGM}$.

*Code availability:* The source code of IMAU-ICE is maintained on GitHub at https://github.com/IMAU-paleo/IMAU-ICE/tree/Last_Glacial_Cycle_PMIP3. The exact version used in this study (including makefiles, compiling scripts, run scripts, config files for all the simulations presented here, and MATLAB scripts for creating the figures) is available at Zenodo (https://doi.org/10.5281/zenodo.7463260). Please note that model simulations cannot be conducted without input files for $CO_2$, climate and initial topography. For more information, contact the corresponding author.

*Data availability:* The simulation output shown in this study is available for a 10 kyr output frequency at Zenodo (https://doi.org/10.5281/zenodo.7463248). Output every 1 kyr as well as additional fields can be requested by contacting the corresponding author.

*Author contributions*. MS has conducted the simulations and written the manuscript. Experimental set-up was created by RW, CB and MS. Model support has been provided by CB and LS. All authors have helped analysing the results and contributed to the text.


*Competing interest*. The authors declare that they have no conflict of interest

*Acknowledgements*. M.D.W. Scherrenberg is funded by the Netherlands Earth System Science Centre (NESSC), which is supported financially by the Ministry of Education, Culture and Science on OCW grant no. 024.002.001. C.J. Berends
was supported by PROTECT. This project has received funding from the European Union's Horizon 2020 research and innovation program under grant agreement no. 869304, PROTECT contribution number [will be assigned upon publication]. The use of supercomputer facilities was sponsored by Dutch Research Council (NWO) Exact and Natural Sciences. Model runs were performed on the Dutch National Supercomputer Snellius. We would like to acknowledge SurfSARA Computing and Networking Services for their support. L. B. Stap is funded by NWO through VENI grant
VI.Veni.202.031. We would finally like to thank the editor and two anonymous reviewers.

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

**Table 1.** The climate forcing from PMIP3 model output that was used in this study. PMIP3-Ensemble represents the mean of COSMOS, IPSL, MIROC and MPI. The global annual temperature (T) and precipitation (P) difference between LGM and PI is shown for each climate model.

| Climate Model | Working Name | Reference | $\Delta T_{LGM}$ (K) | $\Delta P_{LGM}$ (mm/yr) |
|---|---|---|---|---|
| CCSM4 | CCSM | Brady et al., 2013 | 4.8 | 122 |
| CNRM-CM5 | CNRM | Voldoire et al., 2013 | 1.9 | 70 |
| COSMOS-ASO | COSMOS | Budich et al. 2010 | 5.3 | 126 |
| FGOALS-g2 | FGOALS | Zheng & Yu, 2013 | 4.5 | 109 |
| GISS-E2-R | GISS | Ullman et al., 2014 | 4.5 | 102 |
| IPSL-CM5A-LR | IPSL | Dufresne et al., 2013 | 4.6 | 146 |
| MIROC-ESM | MIROC | Sueyoshi et al., 2013 | 4.6 | 112 |
| MPI-ESM-P | MPI | Jungclaus et al., 2012 | 4.4 | 98 |
| MRI-CGCM3 | MRI | Yukimoto et al., 2012 | 4.2 | 141 |
| PMIP3-Ensemble | PMIP3-Ensemble | - | 4.7 | 120 |

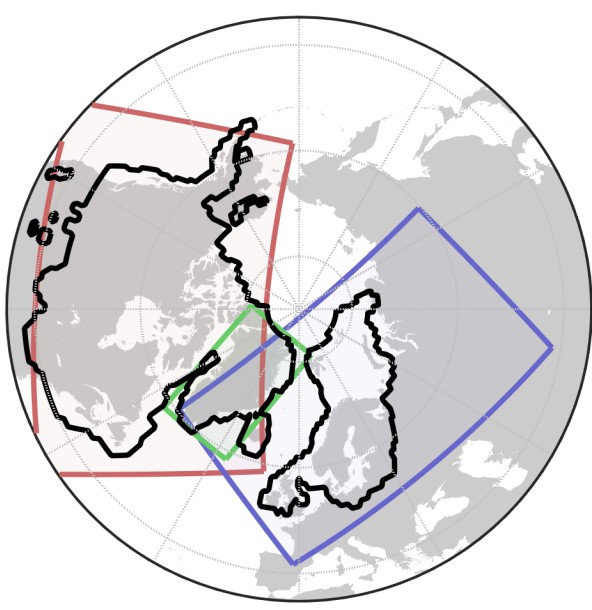

**Figure 1.** The North American (red), Greenland (green) and Eurasian (blue) domains used in the ice sheet model. The LGM extent from Abe-Ouchi et al. (2015) is shown in black. This ice sheet reconstruction is used as a boundary condition in the PMIP3 climate model simulations.

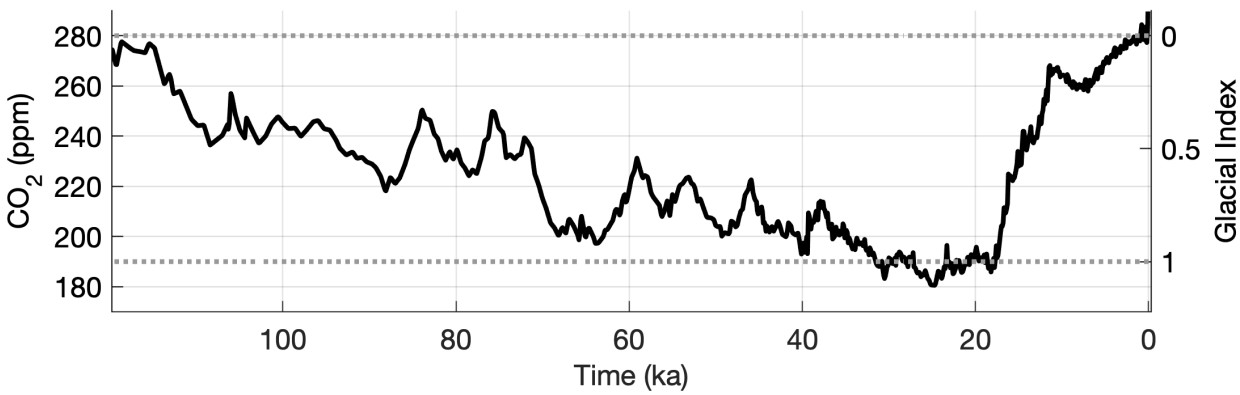

**Figure 2.** Atmospheric $CO_2$ concentrations during the LGC from Bereiter et al. (2015) and the corresponding glacial index values. A glacial index value of 1 represents 'full glacial conditions' corresponding to an LGM climate. A glacial index value of 0, or 'full interglacial conditions', represents a PI climate.

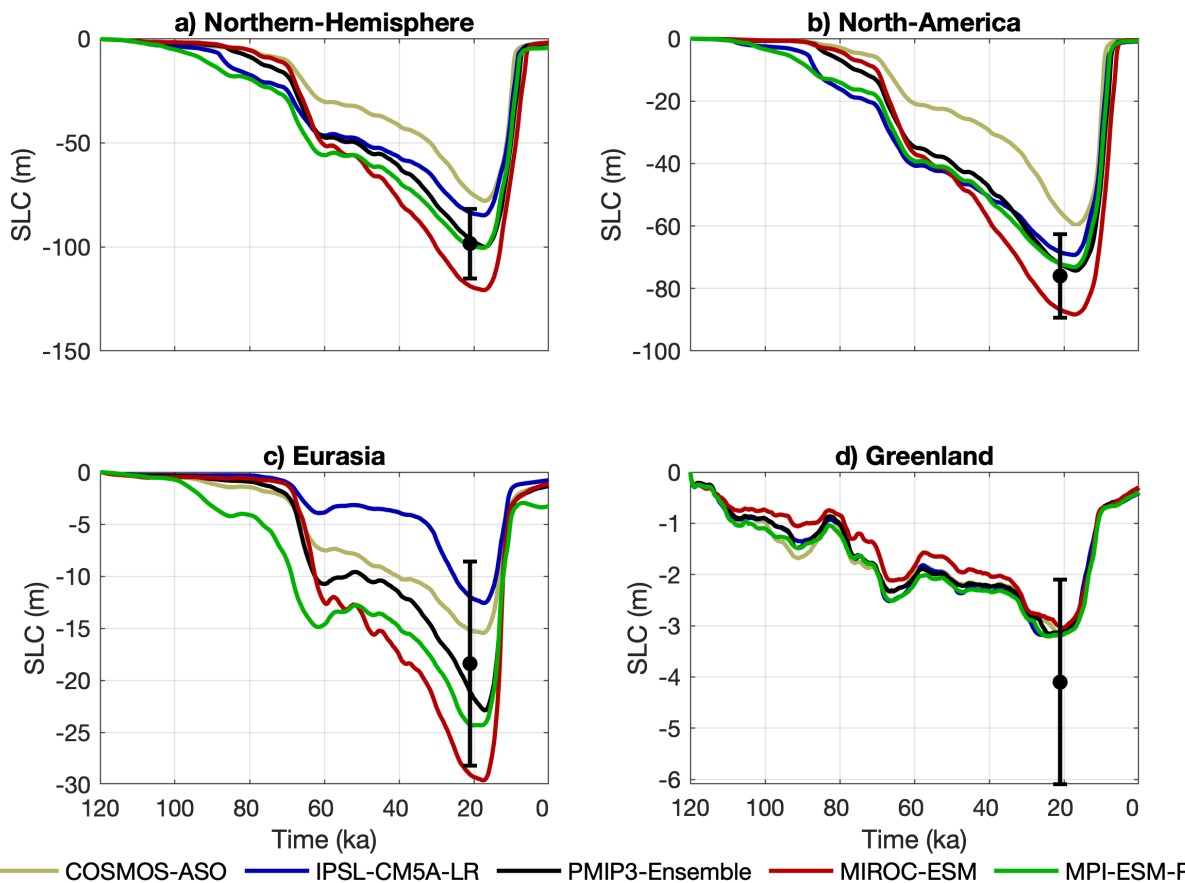

**Figure 3.** Sea level contribution of the Northern Hemisphere ice sheets using the climate matrix method and the uncertainty range of LGM ice volume from Simms et al. (2019). Each simulation was forced with a climate obtained from a member of the PMIP3 ensemble.

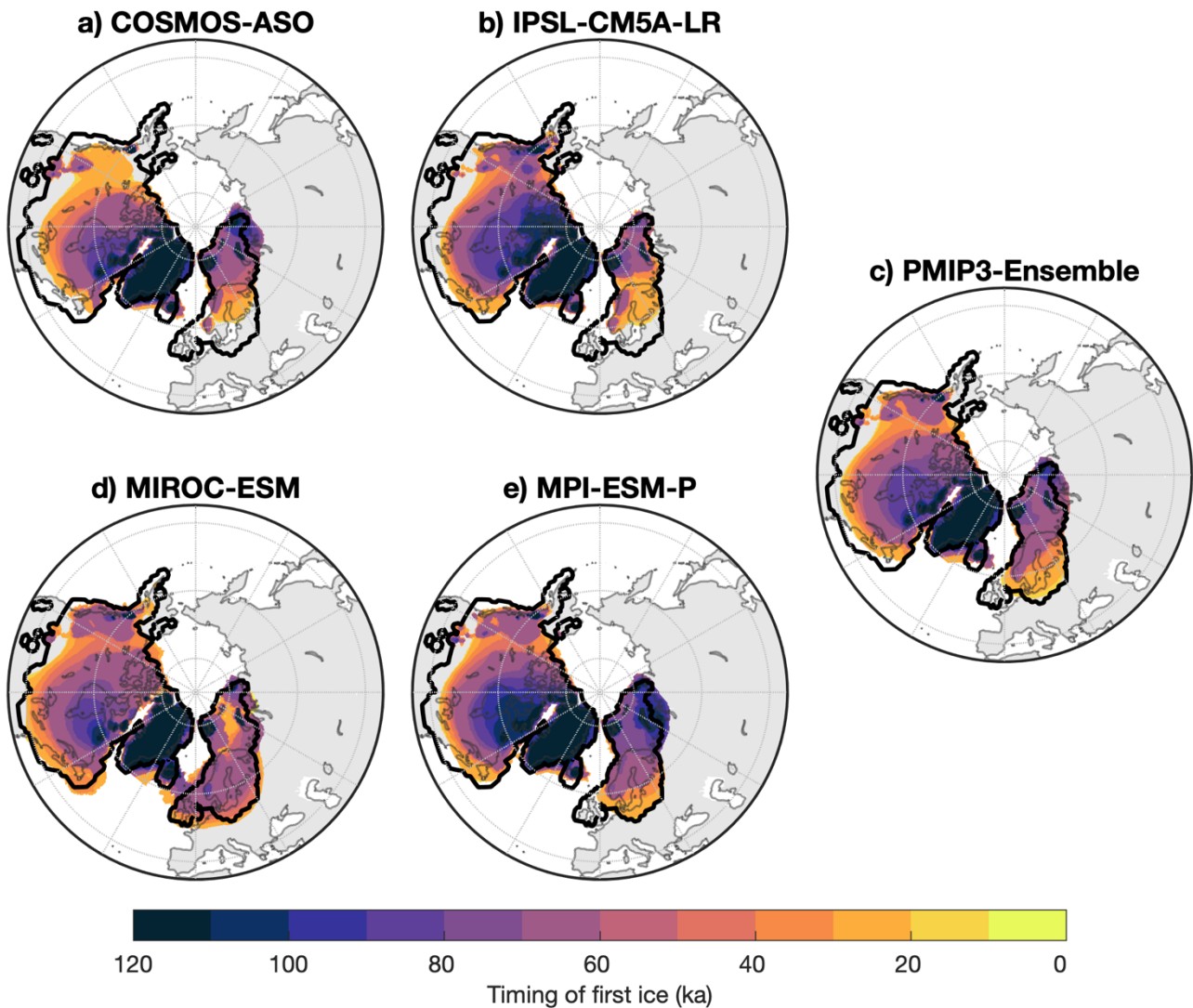

**Figure 4.** The time at which a region was first covered by ice during the LGC. Darker colours represent earlier inception. Lighter colour regions were covered by ice later. The black contour depicts the ice extent reconstruction Abe-Ouchi et al. (2015) used in the GCM simulations. Each simulation was forced by GCM output from PMIP3 interpolated using the climate matrix method.

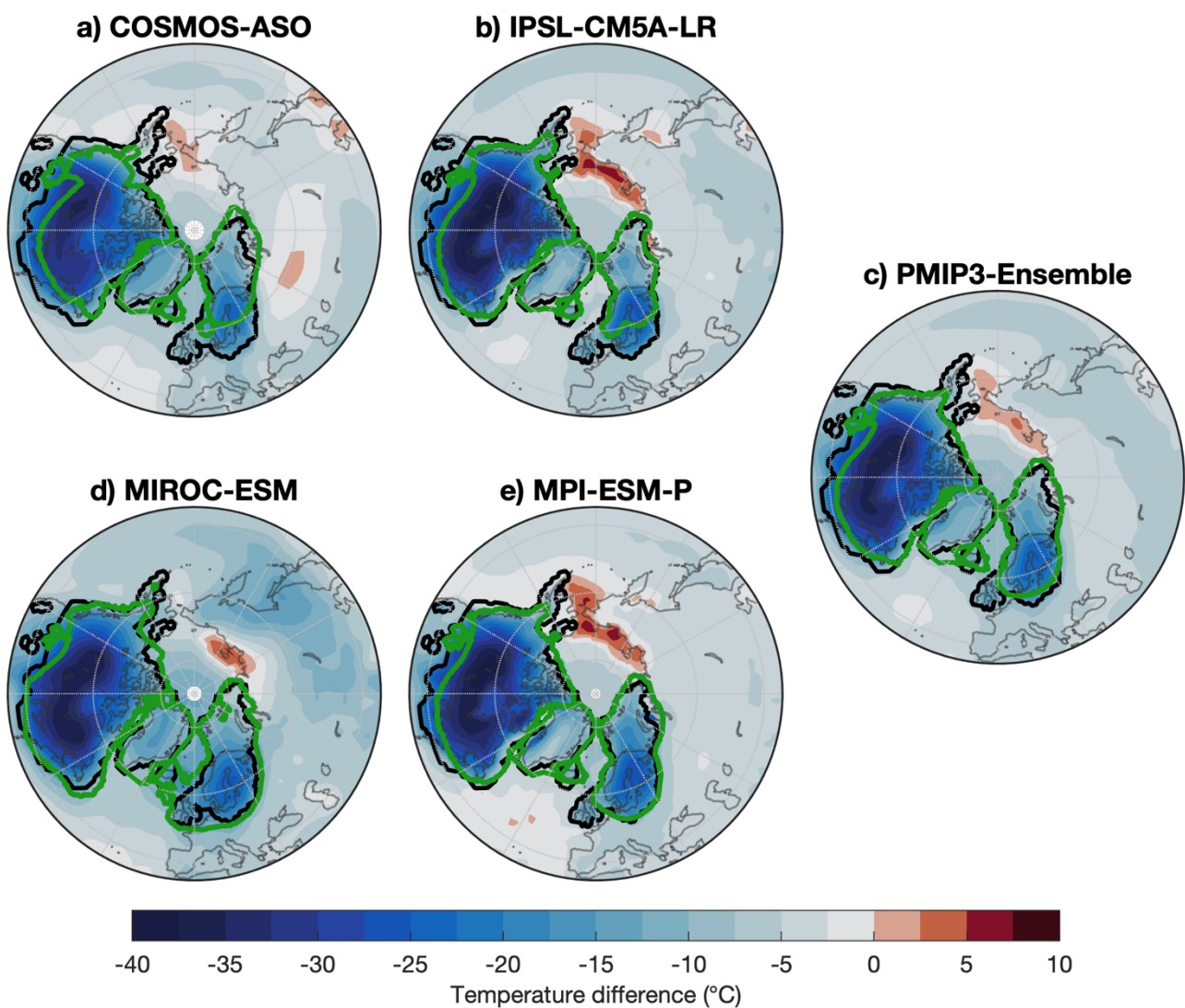

**Figure 5.** Summer (JJA) temperature differences between LGM and PI. IPSL and MPI have lower temperatures in Arctic Canada compared to the other GCM climates, which results in larger rates of ice growth rate at the onset of the LGC. Black contours represent the LGM extent of the Abe-Ouchi et al., 2015 ice sheets. The LGM extent of the ice sheet model simulations is shown in green.

740

745

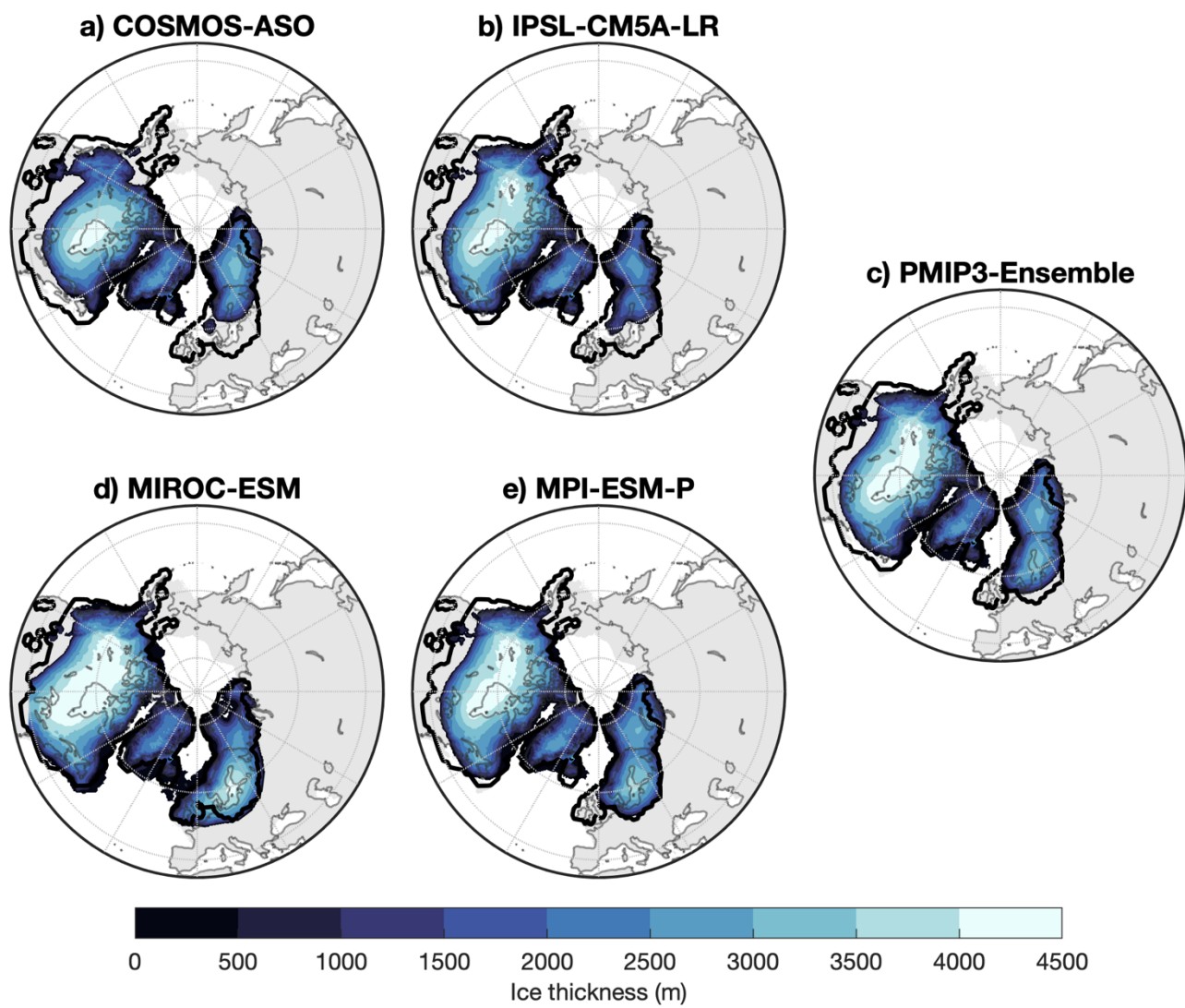

**Figure 6.** Ice thickness using forcing obtained from members of the PMIP3 ensemble. The ice extent by Abe-Ouchi et al. (2015) is shown as a black contour.

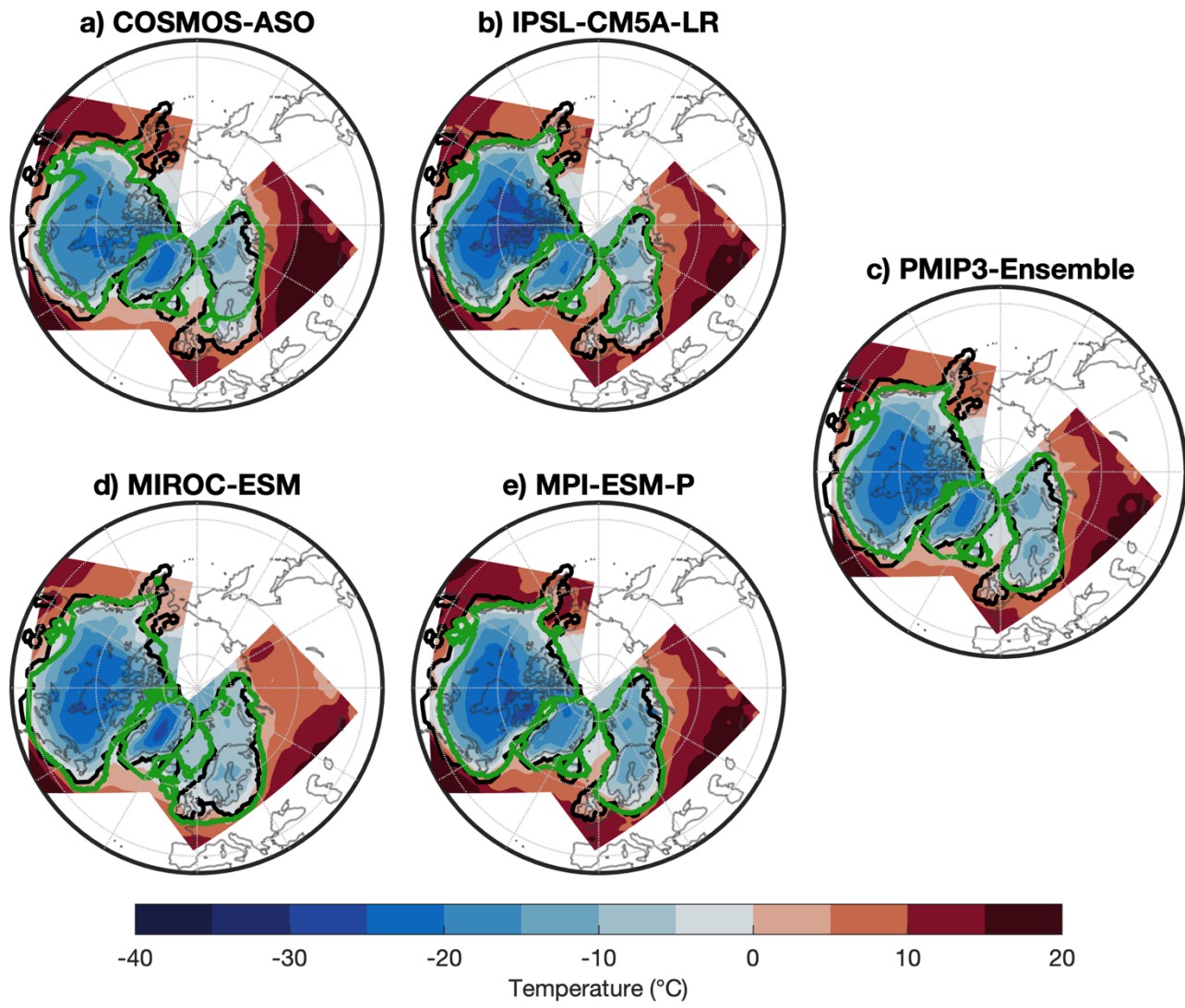

**Figure 7.** LGM summer temperature as it is applied to the ice sheet model. The green contours indicate the extent of the ice sheets that resulted from the GCM forcing. The black contours show the extent of the reconstruction by Abe-Ouchi et al., 2015.

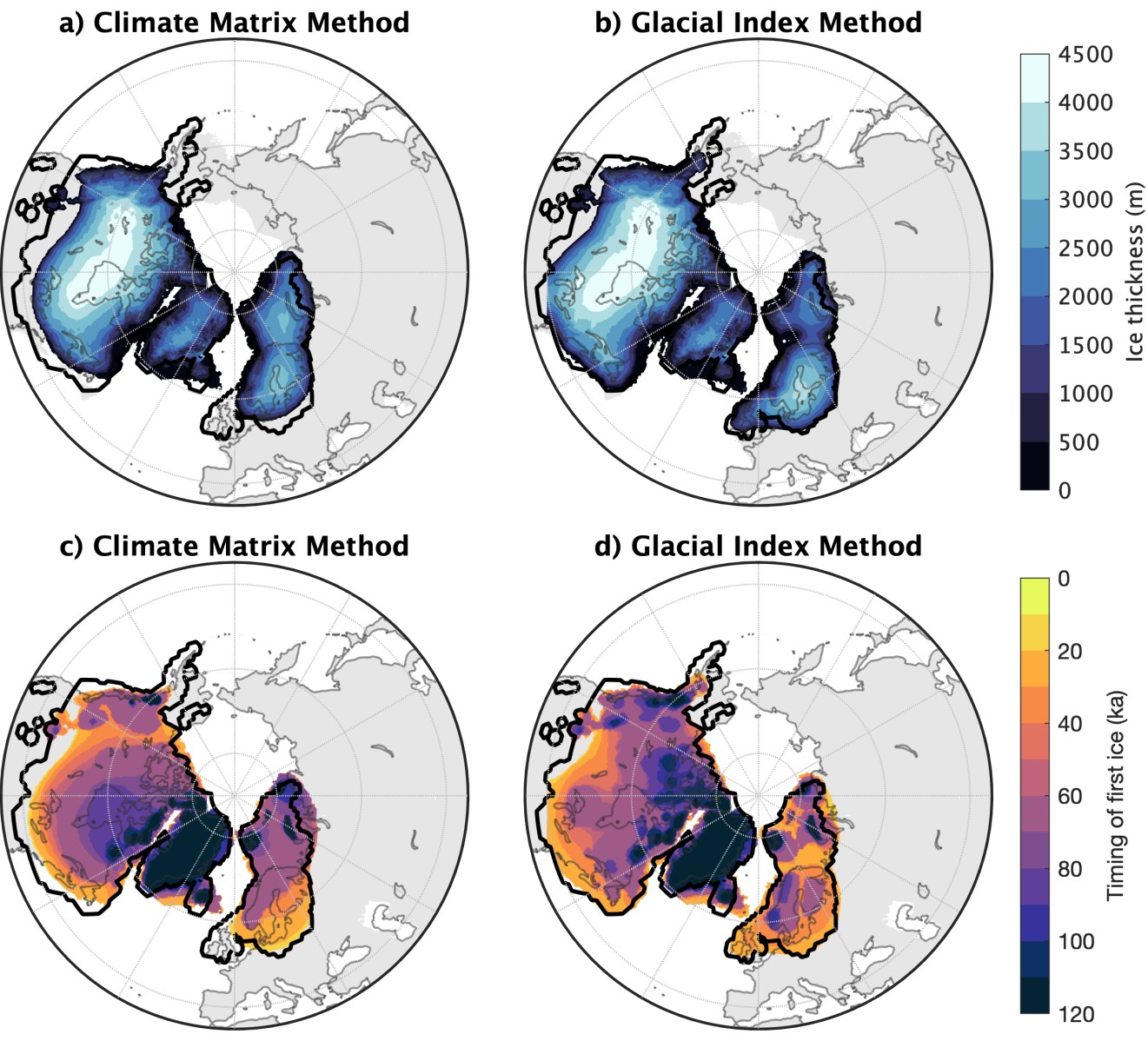

760

**Figure 8.** Ice thickness (a,b) and timing of first ice (c,d) of the climate matrix (a,c) and glacial index (b,d) methods. The black contour represents the extent of the ice reconstruction by Abe-Ouchi et al. (2015).

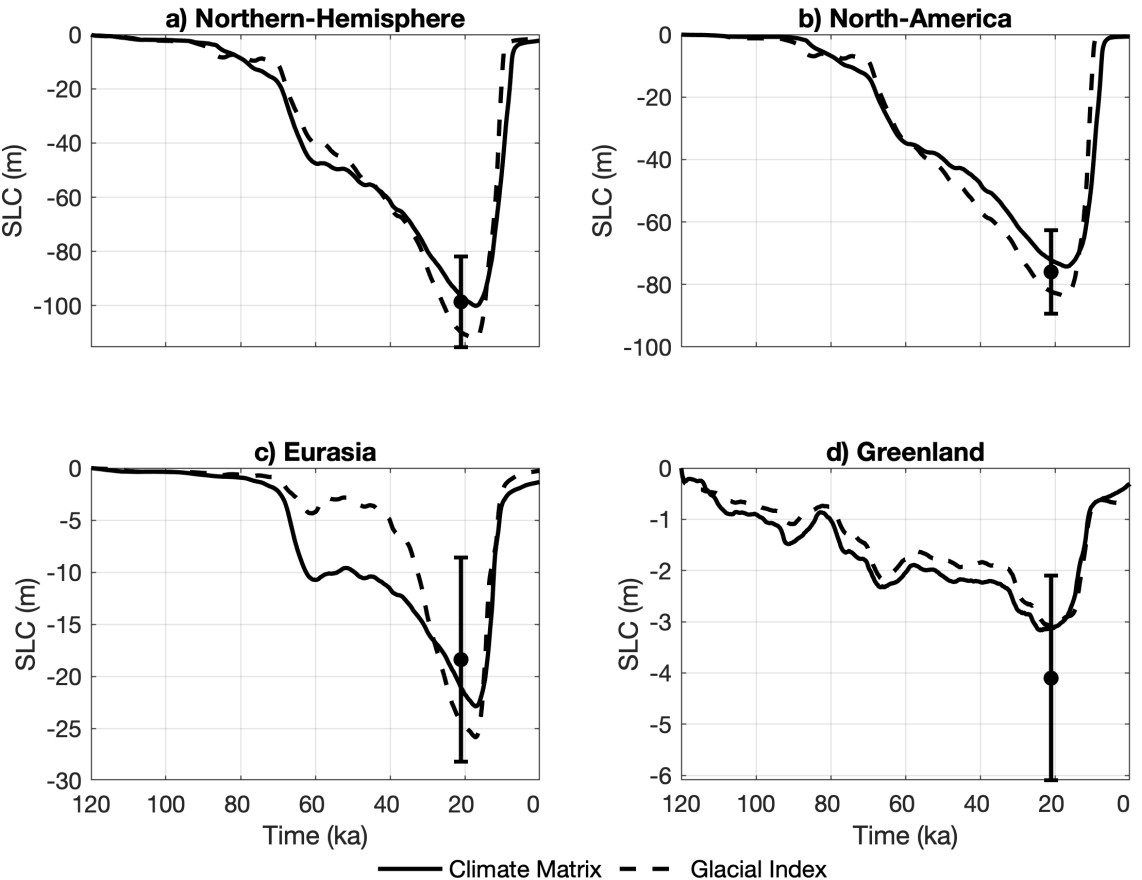

**Figure 9.** Sea level contribution during the LGC for simulations forced by PMIP3-Ensemble using either the glacial index or climate matrix method. The errorbars indicate the sea level contribution by Simms et al., 2019.

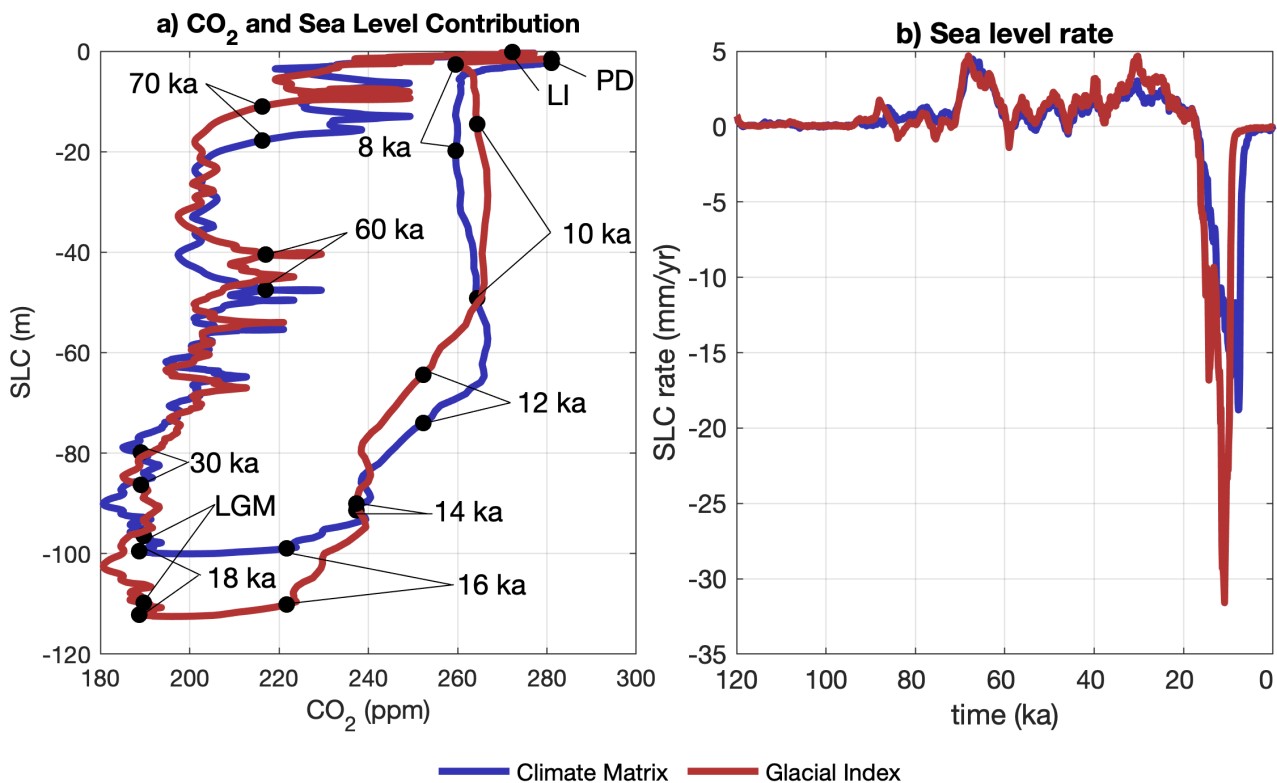

**Figure 10.** Sea level contribution plotted against CO₂ concentrations (a). And (b), The sea level contribution rate during the LGC. LI indicates last interglacial; PD indicates present day.

770