# Peer review of "Modelling feedbacks between the Northern-Hemisphere ice sheets and climate during the last glacial cycle"

_Climate of the Past, 2022_

## Author Comment (AC1)

**Reply to Reviewer 1:**

First of all, we would like to thank the reviewer 1 for their insightful comments. In this document we address the responses to the reviewers' comments. Comments by the reviewer are listed in bold, the authors reply in regular font type.

**General comments**

**The authors define as a matrix method a climate interpolation between only two snapshots. It is in fact a very minimal version of the matrix. I believe that a proper matrix method should use more snapshots (Pollard, 2010; Abe-Ouchi et al., 2013; Ladant et al., 2014). With only two snapshots your ice sheet evolution can be very much constrained by the extent of the reconstructed ice sheet at the LGM. Perhaps you should replace the term "matrix" here or better list the limitations of the approach followed here.**

**For example, an interesting feature of the matrix method is to cover a range of possible ice sheet extent, CO2 and insolation. Here you have a very minimal version of the matrix. Ice sheet extent and CO2 are both extremes in the matrix. But much lower insolation (or larger) is not really explored (while it could be of importance). There are only few GCM simulations of MIS3 or MIS4 but perhaps you could have used the PMIP-lig127k experiment to increase the size of your matrix? You should add a discussion on what would be the benefit of having other snapshots covering the last glacial cycle.**

In our current set-up our temperature and precipitation forcing is obtained by interpolating between two climate snapshots: LGM and PI. These two snapshots represent the cold and warm climate respectively. While this is a limited number of snapshots, we believe that this can still be considered as a climate matrix method.

The defining difference between the matrix and index method is that only the matrix method includes climate feedbacks. Temperature-albedo and precipitation-topography interactions are only included in the climate matrix method. This can be conducted with only two snapshots. However, we do agree with the reviewer that additional snapshots could benefit out simulations, especially to capture feedbacks between insolation forcing, ice sheet topography and atmospheric circulation. This may help us to better capture the two-way effect of atmospheric circulation on ice growth and melt and would possibly improve the temporal evolution.

In this paper we have chosen a range of different GCM simulations over the use of many different snapshots. We, and other researchers such as Niu et al., (2019) and Adler & Hostetler (2019) have found substantial differences in ice sheets forced by different GCM simulations. There is a large uncertainty in temperature and precipitation at LGM, which is why we chose to use the PMIP3 ensemble instead of only using one single GCM.

However, the drawback to this method is that we are limited by the GCM snapshots – or time slices – available.

PMIP-lig127k is available for PMIP4, but is unfortunately unavailable for PMIP3. We chose PMIP3 instead of PMIP4 for a number of reasons. When we started our preliminary experiments, PMIP4 was not yet finished. Using PMIP3 also allowed us to make comparisons with older literature, such as Niu et al., 2019 and Adler & Hostetler 2019. Furthermore, an intercomparison paper by Kageyama et al., 2021 stated that PMIP3 is not fundamentally different from PMIP4. Both phases show large ranges in LGM temperature and precipitation, so we expect a large range in ice sheets with PMIP4 as well. An explanation why we used PMIP3 instead of PMIP4 will be added to the manuscript.

Nevertheless, while the number of snapshots is limited it can still be viewed as a climate matrix since it includes ice-sheet climate interactions. Though, yes, it is quite a limited number of snapshots. We will add this to the discussion. The references stated by the reviewers contain other work conducted using the climate matrix method. These will be added in the introduction and discussion sections.

**Perhaps some of the discrepancy regarding the temporal evolution of the ice sheets through the last glacial cycle?**

As stated by the reviewer, there are differences in the modelled and reconstructed evolution of global mean sea level throughout the last glacial cycle. The largest discrepancies are found during the inception phase where we are unable to obtain enough ice throughout MIS5. LGM volume and extent as well as the deglaciation agree well with reconstructions.

There are two main reasons why the model does not have a good agreement with the temporal evolution during the last glacial cycle. First of all, we believe the main reason is due to a too weak influence of insolation on ice evolution. In our model, insolation affects two processes. First, temperature is interpolated with respect to $CO_2$ and absorbed insolation. The absorbed insolation is calculated using albedo and insolation. Secondly, insolation is used to derive the albedo in our SMB model. However, the effect of insolation on the ice evolution is weak. The insolation minima reached during MIS5 may have favoured ice growth, which is currently not well captured. Ideally, we would have used climate snapshots that contain different orbital parameters, but these are not available for the full PMIP3 ensemble.

The second reason for discrepancy between reconstructions and our ice sheet simulation is that we focused on tuning the model towards LGM. As a result, we obtained reasonable LGM ice sheets and deglaciation, but at a cost of the quality of the pre-LGM ice sheet.

We will extent the discussion section to address these issues concerning the temporal evolution of the ice sheet.

**I think that all this should be better acknowledged somewhere in the manuscript.**

**- model of intermediate complexity.**

**There is an wide literature on ice sheet – climate coupling using climate model of intermediate complexity, which is very relevant for the topic addressed here. At present in the manuscript only GCMs (forced or coupled) are discussed. Elements on intermediate complexity model approaches should appear in the introduction as well as in the discussion. Although simpler than GCMs in their physics, the intermediate complexity model can explicitly represent ice sheet climate interactions instead of using parametrisation as in here.**

In the manuscript we did not explicitly touch upon intermediate complexity models in the introduction, except for a reference to Ganopolksi et al., 2010. Intermediate complexity models were not addressed in the discussion either.

We agree with the reviewer that intermediate complexity models are indeed a useful addition to simulate ice-sheet – climate interactions for specific problems, but they also have limitations for instance related to precipitation changes. In the end transient GCM are required, though these are not feasible yet on the time-scales of glacial cycles. For the time being we have to work with simplifications such as a matrix-method or models of intermediate complexity.

Therefore, we will make a clearer statement in the introduction that intermediate complexity models can be used to more explicitly resolve climate-ice sheet feedback at a reduced computational cost compared to GCM's.

**- ice sheet – climate feedbacks.**

**There is an oversimplification in the manuscript when it is stated that the climate "matrix" is able to take into account the ice sheet – climate feedbacks. From my understanding the climate matrix interpolation method improves the glacial index method on two points: a simple way to account for different albedos / insolation and a more clever way to parametrise the precipitation correction. Ice sheet – climate feedbacks are more complex. For instance the geometry of the ice sheet will affect the precipitation/temperature patterns not only due to height changes but also due to atmospheric circulation changes. Here only two geometries are used (LGM and PI) and intermediate ice sheet sizes or alternative geometries cannot be taken into account. Also, an other major ice sheet – climate feedbacks is the melt freshwater release into the ocean and its impact on oceanic circulation. This is completely ignored by the climate matrix method (also because the PMIP GCMs do not explore this neither).**

**That being said, I think that the climate matrix as presented is a nice alternative to the index method but the authors might rephrase the manuscript in some places (e.g. l.18, section 3.2, l.284).**

The climate matrix method is able to calculate a temperature-albedo feedback and precipitation-topography feedback, however there are indeed several limitations regarding ice-climate interactions.

In the manuscript, we stated that climate matrix method is unable to simulate some tipping / threshold behaviour in the climate system, such as the closure of the Arctic Archipelago gateway. However, we did not state that our method is unable to resolve feedbacks involving atmospheric circulation changes or oceanic changes. To better resolve this using the climate matrix method, this would require either many snapshots, which are not available for the full PMIP3 ensemble, or a transient climate model simulation – which we are trying to avoid due to computational time.

We agree with the reviewer and will add a more thorough discussion on the limitations of our method regarding the interaction between atmospheric and ocean circulations. We will refer to, amongst other, work conducted by Liakka et al., 2016 and Löfverström et al., 2017 to discuss interactions involving atmospheric circulation. Furthermore, we will explicitly state that the climate matrix method is meant to be an alternative to the glacial index method, and it should not be viewed as an alternative to climate models.

**- Ice sheet – ocean interactions.**

**There is very little info to the way ocean forcing is treated here (only a reference to de Boer paper). And the authors do not invoke oceanic circulation changes in shaping the last glacial cycle while they were probably critical. Do you have noticeable differences in terms of sub-shelf melt through the glacial cycle? Does the model is sensitive to these changes? A dedicated section could be welcome.**

Ocean-ice interactions are limited in our current set-up. We use a very simple basal melt model. To calculate melt we use the parameterization by (Martin et al. 2011). This parameterization uses globally uniform ocean temperatures to calculate basal melt, for which we used the approach by De Boer et al., 2013. As a result, our basal melt model is of limited value. These temperature fields were not taken from a GCM and do not capture spatial temperature patterns. Neither do we include any interactions between the ice sheet and ocean circulation.

Our simplistic approach to sub-shelf melt rates may have had consequences to the evolution of the ice sheets. In Eurasia, the northern and western margin of the ice sheet are often marine throughout the last glacial cycle. In North America, the margin is marine in the Canadian Arctic. Though the Southern margin only is in contact with water during the deglaciation. For Greenland, the entire ice sheet might become ocean terminating. These limitations will be addressed in the discussion, and we will more explicitly mention the simplicity of our basal melt model in the method section.

**- Grids.**

**The different ice sheets are fully disconnected here, while there could be some interactions between the Innuitian ice sheet with the Greenland ice sheet. Also**

**there could be an ice shelf in the Baffin Bay affecting the North American ice dynamics. Have you tried to include such interactions? It might deserve a discussion at some point.**

In our current model-set up, the North America, Eurasian and Greenland ice sheets are modelled in separate domains. Therefore, except for changes in sea level the ice sheets do not interact. During the last glacial cycle, the North American and Greenland ice sheets were likely attached, which is not possible using our current set-up.

However, since we do not have (sea) ice–ocean feedbacks, we do not expect that a shelf between North America and Greenland will make a significant difference in the evolution of the ice sheets. Furthermore, the effect of ice dynamics on a separate Greenland and North American ice sheet should be minor. We will add a remark on discrepancy in the discussion.

**A minor thing: the North American grid seem too small, for example MIROC-ESM is cut in the South. It is not too bad here since, except for MIROC-ESM, all your climate forcing produces smaller ice sheet than the reconstructions.**

In figure B1 and figure 6 ice thickness is shown. Here we can see that the ice sheet produced using MIROC-ESM temperature and precipitation forcing slightly exceeds our modelled domain. The ice sheet touches the southern boundary of our ice model. Similarly, figure B1 shows that ice sheets forced by GISS and FGOALS touch the domain edges in Eurasia.

We do not simulate the entire globe using our method. Instead, to save on computational time, we simulate each ice sheet in a small domain that is big enough to support the LGM reconstruction. When the climate forcing results into an ice sheet bigger than the LGM reconstruction, the ice sheet may touch the edge of the domain. So, we may slightly underestimate the size of the LGM ice sheet for MIROC. We will address this in the results section of the manuscript. If the ice sheet grows outside the domain it is considered to be too large anyhow.

**- Minimal version of albedo.**

**The background albedos are tunable but close to what we expect but how the two critical parameters (set here to 15 and 0.015) have been selected (Eq. 2, l. 321)? Could you show a map of the albedo from RACMO and from the parametrisation to see how the formulation performs? Such a comparison with a timeseries for a grid point in the ablation area could be also very nice.**

Our albedo parameterization is based on Bintanja et al., 2002. It first calculates a background albedo. This is 0.1 for water, 0.2 for land, 0.5 for bare ice. Snow is added on top of land or ice, which can increase the albedo to a maximum of 0.85. The final albedo is capped between this background albedo and the snow albedo. This parameterization, along with our entire SMB model, was used as part of the intercomparison project

GrSMBMIP (Fettweis et al., 2020). This will be more explicitly mentioned in the manuscript.

Secondly, we found a small mistake was made when writing this equation in the manuscript. In the manuscript, MeltPreviousYr had ended up in the exponent, but should have ended up outside. This will be fixed in the manuscript. As it is a writing error, it has no effect on the model results.

Wrong version:

$$\alpha_{surface}(x,y,m) = \alpha_{snow} - \left(\alpha_{snow} - \alpha_{background}\right)e^{-15\,FirnDepth(m-1,x,y)-0.015\,MeltPreviousYr(x,y)}$$

Correct version:

$$\alpha_{surface}(x,y,m) = \alpha_{snow} - \left(\alpha_{snow} - \alpha_{background}\right)e^{-15\,FirnDepth(x,y,m-1)} - 0.015\,\mathbf{MeltPreviousYr(x,y)}$$

**- Climate matrix parameters.**

**Please justify the use of different methodology / weighing factors for the different ice sheets (l. 160-161 and l. 415-420). Explain the choice of the different weighing factors (l. 415-420).**

We apply slightly different methodologies for Greenland compared to Eurasia and North America. These choices, which follow the approach outlined by Berends et al. (2018,2019), were essentially made due to the small change in the topography and extent of the Greenland ice sheet compared to North America and Eurasia.

First of all, when calculating the albedo feedback in Greenland (l 415-420) we apply a less strong effect of the local albedo towards temperature. This is because Greenland has ice cover in pre-industrial and LGM. Therefore, for most of Greenland, albedo stays roughly the same throughout the last glacial cycle. The biggest change in albedo occurs as ice shelves grow around the island. It therefore makes more sense to let temperature be controlled by the total change in albedo instead of the local albedo changes.

Instead, both the Eurasian and North American ice sheets are ice-free during pre-industrial. The albedo changes throughout the entire ice sheet therefore allowing us to use local effects of albedo.

Similarly, we also used a slightly different method to account for orographic forcing of precipitation. In Greenland, the change in topography is relatively small compared to the other ice sheets. Therefore, we use a Clausius Clapeyron relation to correct for the change in precipitation due to changes in temperature. In Eurasia and North America, the topography changes much more dramatically throughout the glacial cycle.

Topography has a strong effect on precipitation, as precipitation is higher when air moves upslope and lower when air moves downslope. Therefore, we correct for topography using the Roe and Lindzen (2001) model instead.

These reasons for these slightly different methods will be added to the manuscript.

**- Ice sheets too thick.**

**Can you comment on the respective role of climate forcing and basal drag to explain this bias?**

Our simulated ice sheets are relatively thick in our simulations compared to reconstructions. While we are able to obtain a good LGM ice volume, the extent is slightly too small.

The ice sheet could be too thick for a number of reasons. Ice dynamical processes, amongst others basal drag, could be a main reason for the ice sheet being too thick, because not enough ice is being transported from the interior to the ice sheet towards the margins. Basal friction is not well constrained, and we use the same parameterization for each ice sheet. In this parameterization, friction increases with the height of the bedrock. Therefore, regions with low bedrock elevations, such as ice streams, have low friction. Basal friction in ice shelves is negligible and set to 0. However, this parametrization has some issues. For example, observations are not used for basal friction.  We do not simulate sediment thickness, which may substantially impact basal sliding in Eurasia and North America.

Climate forcing may have led to a large ice thickness, but we believe it is better constrained compared to the ice dynamics. The large ice thickness is found in simulations forced by most PMIP3 GCM's, except those yielding very small ice sheets. Once the ice sheet approaches the LGM thickness, the precipitation forcing will be close to LGM, making ice plateaus arid throughout the LGC.

We will add a sentence to the manuscript suggesting that the ice sheet being too thick could be a problem with ice dynamics, specifically basal sliding.

**Specific comments**

**l. 30-31 and 35: the ice sheets respond on multi-millennial timescales but they can show abrupt changes as well (marine ice sheet instabilities, saddle-collapse,...). Please rephrase here.**

In the manuscript we stated that ice sheets generally respond slowly. While this is generally true, there are processes that are much more abrupt, as suggested by the reviewer. We will change this sentence to not exclude rapid changes in the ice sheet. We will mention that ice sheets generally respond on multi-millennial timescales, but there are processes that can respond on short time-scales. Here we will add references to Gomez et al., 2015 and Brendryen et al., 2020.

**l. 38-39: ice thickness and extent are NOT known over "millions of years"... eventually eustatic sea level is known to a certain degree... Please rephrase.**

In the manuscript we stated that paleo reconstructions contain information on the climate system such as ice sheets over millions of years. This is not true, and we will change the sentence accordingly. The sentence will now only mention that paleo-reconstruction contain information of the climate system before modern observations could be made.

**l. 86: "the LGM"**

This grammar mistake will be fixed.

**l. 130: Even though the albedo and topography changes are expected to be smaller for the Antarctic ice sheet with respect to the Northern Hemisphere ice sheets, changes there can strongly affect the Atlantic meridional overturning circulation (deep water formation), hence Northern Hemisphere climate. Add a sentence on this here.**

The Antarctic ice sheet is not included in our simulations. This is due to Antarctica only having a small effect on the total eustatic sea level changes. However, the Antarctic ice sheet has a major impact on ocean circulation and the carbon cycle. This needs to be mentioned in the manuscript.

We will add a sentence stating that Antarctica has a major impact on the carbon cycle and ocean circulation. In the method's section, we will refer to Adkins et al., 2013 who investigated the effect of Antarctica on the carbon cycle and ocean circulation.

Furthermore, one of the limitations to the climate matrix method is that we do not explicitly simulate interactions with the carbon cycle or ocean circulations. These limitations will be mentioned in the discussion section.

**l. 160: Greenland spatial resolution is higher so the model can see some change in topography. Please justify better why you do not follow the same protocol for this ice sheet.**

Greenland has been simulated at a higher resolution due to a number of reasons. First of all, increasing resolution helps to resolve ice streams and small topography changes better. By increasing the resolution in Greenland from 40 to 20 km, the computational time is increased. However, since the Greenland ice sheet is relatively small, this higher resolution yields a comparatively run-time compared to the North American and Eurasian ice sheets.

We will add explanations on why we used slightly different methods for Greenland compared to North America and Eurasia. This includes why we used the Clausius Clapeyron relation instead of the Roe and Lindzen model, the different weighting factors for temperature in the climate matrix method, and the use of different spatial

resolution. These choices were made due to the small change in extent and topography of the Greenland ice sheets compared to Eurasia and North America.

**l. 177: problem with the sentence.**

This sentence will be fixed.

**l. 215-220: Sub-shelf melt / oceanic biases can be a reason for the Eurasian ice sheet bias?**

The Eurasian ice sheet has a too small ice coverage in Western Europe, especially the British islands. This bias may be caused by a number of different processes. First of all, as stated by the reviewer, there may be too much melt in ice shelves. This prevents ice sheet expansion towards the Western Europe. Secondly, it may be a result of climate forcing – namely a too high LGM temperature or too low precipitation. We obtained ice in the British islands when using MIROC forcing. Thirdly, we do not explicitly simulate atmospheric feedback processes, which may have an effect of the evolution of the Eurasian ice sheet. Ice dynamics could play a role as well. The LGM ice sheet is relatively too thick, while having a slightly too small ice extent. This can likely be attributed to ice dynamics, as perhaps not enough ice is transported from the interior towards the margins.

To summarize, several processes may have led to the discrepancies in Western Europe. Each of these discrepancies will be mentioned in the discussion section.

**l. 225-230: Basal drag vs. precipitation bias?**

This line mentions that the ice sheets simulated with our set-up are relatively thick compared to reconstructions. This may be attributed to ice dynamical processes. The ice sheet has a good LGM ice volume, but a slightly too small extent while being too thick in the interior. This suggest that not enough ice is transported from the interior towards the margins.

Furthermore, the ice sheet is too thick in seven out of nine simulations forced with different PMIP3 models. This indicates that precipitation forcing may not be the main reason for the discrepancy in ice thickness. We will make a clearer statement showing that ice dynamics – and specifically basal drag – may have resulted in a too large ice thickness.

---

## Author Comment (AC2)

**Reply Reviewer 2:**

First of all, we would like to thank the reviewer 2 for his/her insightful comments as well as providing additional literature suggestions. In this document, replies are listed with regular font type, while the reviewers' comments are listed in bold.

**General comments:**

**(i) The discussion section is relatively weak in its current form and could benefit from some revisions. For example, you mention that both the index and the matrix methods are missing certain processes and cannot realistically represent abrupt circulation changes. I agree with this statement, and I would like to see a more thorough discussion on how this shortcoming is (potentially) influencing your results. Here are a few papers that have examined and at least partially explained abrupt changes in the large-scale atmospheric circulation in the last glacial period:**

**https://agupubs.onlinelibrary.wiley.com/doi/full/10.1002/2017GL074274**
**https://journals.ametsoc.org/view/journals/atsc/73/8/jas-d-15-0295.1.xml**

**https://cp.copernicus.org/articles/12/1225/2016/**

We agree with the reviewer and we will add a more thorough discussion in the manuscript on limitations regarding modelling climate feedbacks. In our current discussion we stated that we are unable to simulate threshold behaviour in the climate system with our current set-up, such as the effect of ocean circulation on the closure of the Arctic Archipelago gateway. However, the discussion should also include our method's limitation regarding feedbacks between ice and atmospheric /oceanic circulation.

The papers suggested by the reviewer will be included in the manuscript. With these changes, the discussion will contain remarks on the effects of atmospheric circulation on the North American and Eurasian ice sheet (Liakka et al., 2016), abrupt changes in atmospheric circulation in the North Atlantic during deglaciation (Löfverström et al., 2017), ocean circulation changes due to fresh water influx into the ocean (Otto-Bliesner et al., 2010). The manuscript will also benefit from explicitly mentioning that the climate matrix method should not be viewed as a replacement or improvement for GCM models or intermediate complexity models. Instead, the climate matrix method should be viewed as an alternative to the glacial index method which has the effect that the ice sheet evolution influences the dynamics rather than acting as a passive response to a climate forcing as in a glacial index method.

**(ii) The manuscript could benefit from including a supplementary document that shows the simulated model climates (pre-industrial and LGM), and at least a few snapshots of the ice sheets prior to the LGM. I would suggest showing the ice sheets every 30 kyrs or so. It could also be good to compare these fields with some proxy data to better understand the quality of the simulation and what errors the different methods introduce.**

The manuscript currently shows maps of the last glacial maximum (21 ka) ice thickness. It also includes maps of the evolution of ice extent. These maps show the time at which ice

accumulates for the first time in a region. However, we can add maps to the supplementary information showing ice thickness between the glacial index and climate matrix method for every 20 thousand years. This should help to intercompare the ice dome shape and sizes, which differ substantially between the climate matrix and glacial index method. Similarly, we can add GCM LGM and PI temperatures in the supplementary information.

Regarding the temporal evolution of the ice sheets, our current model shows considerable discrepancies in ice volume during MIS5. This can most likely be attributed to a too weak effect of insolation on ice growth. During MIS5, summer insolation in the Arctic reaches a minimum, which currently has a minimal effect on ice growth. Insolation is included in our SMB model as it increases ablation. The temperature forcing in the climate matrix method depends slightly on insolation. Temperature is interpolated with respect to $CO_2$ and annually averaged absorbed insolation. Absorbed insolation is calculated using insolation and albedo. However, the net effect of insolation on ice evolution is limited. This may explain the largest discrepancy in ice evolution, including the slow inception and the relatively small volume at 60 ka. This limitation will be elaborated in the discussion section of the manuscript.

In this work we focused on obtaining a realistic LGM ice sheet, rather than having an optimal transient simulation. The model is tuned to obtain good ice volumes at the LGM and at the end of the deglaciation. However, we did not tune it to obtain a good ice volume throughout the LGC. We will emphasis this point better in the manuscript.

**(iii) What is the reason for using PMIP3/CMIP5 models instead of the updated PMIP4/CMIP6 models (documented here: https://cp.copernicus.org/articles/17/1065/2021/)? Do you have any reasons to assume that the results are robust/not robust across PMIP generations? The paper by Kageyama et al (2021) should be cited no matter what as it documents similarities and differences between the LGM simulations in the older PMIP3 (used here) and the newer PMIP4 models.**

In our manuscript, we have used climate forcing from the paleoclimate modelling intercomparison project phase III (PMIP3). The successor of PMIP3, PMIP4, currently has several LGM and PI simulations.

However, we did not use PMIP4 for several reasons. First of all, when we started conducting preliminary experiments for this paper, PMIP4 was not yet finished, so we used PMIP3 instead. An additional benefit to this was that we could make comparisons to earlier conducted research such as the ice sheet simulations by Niu et al., 2019 and Adler & Hostetler 2019. In addition, Kageyama et al., 2021 stated that PMIP3 is not fundamentally different from PMIP4. A wide range in LGM temperature and precipitation is both found in PMIP3 and PMIP4, so we would expect to find a large range in ice sheets as well. Hence, we preferred the possibility of a more direct comparison to Niu et al and Adler and Hostetler and sticked to PMIP3. We will explain this in the manuscript with reference to the Kageyama et al. 2021 paper.

**(iv) This study is suggesting that the Eurasian Ice Sheet was at maximum extent/volume around 60 ka. This is not captured in your results at all. Is this a result of the lack of "realistic" circulation changes?**

**https://cp.copernicus.org/articles/9/2365/2013/**

In our model, the Eurasian ice sheet reaches maximum volume at LGM, instead of the 60 ka suggested by reconstructions. We believe that this discrepancy, and slow ice inception during MIS5, is partly due to a too weak influence of insolation on temperature and SMB.

In both our glacial index and climate matrix methods, temperature change is mostly driven by $CO_2$ changes. While insolation is used both in the SMB model as well as the albedo feedback, it's overall influence on ice evolution is limited.

During MIS5, summer insolation in the Artic reaches a minimum, while $CO_2$ is still relatively high. Since temperature is mostly driven by changes in $CO_2$ we are unable to capture the fast growth in ice during MIS5. Faster inception, as well as a stronger dependence on insolation may help to reach a maximum Eurasian volume at 60 ka.

As stated by the reviewer, another reason for the discrepancy could involve atmospheric circulation changes. The topography of the North American ice sheet may affect the size of the Eurasian ice sheet, as stated by e.g., Liakka et al., 2016. This interaction is not simulated using our method.

These discrepancies, as well as these aforementioned reasons will be added in the method and discussion section.

**Line comments:**

**Line 1: The title is a bit misleading since you didn't really study coupled interactions between ice sheets and climate. Consider changing the title to be a bit better suited for your study**

The title of our manuscript: *Interactions between the Northern-Hemisphere ice sheets and climate during the last glacial cycle* may imply that we used a transient climate model or reconstructions / observations.

Instead, we will change the title to: *Modelling feedbacks between the Northern-Hemisphere ice sheet and climate during the last glacial cycle*. This title should reflect that we have conducted a modelling study and investigated some feedback processes between ice sheet in the Northern Hemisphere and the climate system.

**Line 11: pre-industrial should not be capitalized**

**Line 12: computationally unfeasible**

**Line 22: exceeds --> exceed**

**Line 23: Specify that this is referring to ice sheet volume**

**Line 40: Rearrange the sentence to increase readability**

The five comments listed above state some small grammar and spelling mistakes, each of which will be fixed.

**Line 42: There are newer references that are more appropriate here:**

**https://www.nature.com/articles/s41586-020-2617-x**

**https://cp.copernicus.org/articles/18/1883/2022/**

The references suggested by the reviewer refer to global temperature reconstructions at LGM. In the manuscript, we used a relatively old reference (Annan & Hargreaves et al., 2013). The reviewer suggested some references to research that has been conducted more recently, which will replace the 2013 reference. We will follow this suggestion.

**Lines 50-54: These types of sentence constructions are difficult to read. Please consider reformulating in a more general way that is not including both cases.**

This comment refers to a line that can be read in two different ways: e.g., the albedo increases (decreases) with decreasing (increasing) temperature.

These types of sentences, while concise, can be difficult to read. We will change it accordingly to improve readability. A sentence employing a similar technique in the discussion section will be changed as well.

**Line 84: A similar technique was employed in:**

**https://gmd.copernicus.org/articles/7/1183/2014/**

In this line we state different methods that have been used in the past to create climate forcing without a transient GCM model.

The paper mentioned by the reviewer refers to a paper by Fyke et al., 2014. They have interpolated LGM, mid-Holocene and PI surface mass balances to simulate the last glacial cycle. This research will be added to this sentence.

**Lines 110-113: This preamble is not necessary and should be deleted**

This preamble for the method section is going to be deleted

**Lines 115-125: There are several abbreviations here that are not defined: IMAU-ICE; SIA/SSA; PISM; CISM**

This part of the method sections contained some abbreviations that were not defined. We have added definitions for SIA/SSA, PISM and CISM. IMAU-ICE is not technically an abbreviation and cannot be defined. This ice sheet model was developed at our institute (IMAU), hence the name IMAU-ICE.

**Line 123: The abbreviation ELRA is only used here and should be omitted**

An abbreviation for ELRA is indeed not necessary and will be removed.

**Lines 160-161: Did you test the sensitivity of this assumption?**

This line refers to a correction that we apply transiently to precipitation. In Greenland we use the Clausius Clapeyron relation to correct precipitation for changes in temperature. We believe this is justified because the shape and elevation of Greenland only experiences relatively small changes throughout the last glacial cycle. However, the topography of the North American and Eurasian ice sheets changes substantially throughout the last glacial cycle. Therefore, we need to apply a correction to account for changes in topography. As precipitation is enhanced up slopes and is decreased down slope. Therefore, we use the Roe and Lindzen (2001) model. This model uses wind and surface slope to correct precipitation changes for topography change. The reason behind this choice will be added to the method section.

However, we did not test the sensitivity between the Clausius Clapeyron and the Roe and Lindzen model.

**Lines 174-174: How is the planetary albedo calculated? Clouds will affect the amount of insolation at the surface...**

A surface albedo is calculated in the ice sheet model. This albedo model first applies a background albedo (land, sea or bare ice) and adds a snow layer on top. This is sufficient for our set-up. The SMB scheme does not include cloudiness explicitly.

We will include a sentence in the manuscript stating that we specifically use surface albedo and did not take cloud coverage into account.

**Lines 180-181: This preamble is not necessary and should be deleted**

The results section contained a one sentence preamble to state what will be discussed in the section. This preamble will be deleted.

**Lines 196-197: I assume that the PI simulations included the observed ice caps on these islands. Thus, the climate is already primed (through albedo effects) to grow ice there. Perhaps a small point, but potentially important to comment on here and in the discussion section.**

During pre-industrial some regions in North America and Eurasia have ice caps. This includes the islands surrounding the Barents Sea (e.g., Nova Zembla, Svalbard) as well as Iceland and the Canadian Arctic Archipelago. We start our simulations with no ice in the North American and Eurasian domains. As a result, these artic regions are already favourable for ice growth and near pre-industrial temperatures are enough to incept ice sheets. So, we agree with the reviewer.

However, the main goal of this sentence is to help explain the accompanying figure; figure 4. Though we will add a small remark that these regions were close to full glaciation during the pre-industrial period. Therefore, it is reasonable that these regions are the first to incept ice.

**Table 1: The LGM simulation with CNRM-CM5 didn't include the ice sheets! Therefore, you may wanna exclude that model from the study. See point 19 here:**

**https://www.umr-cnrm.fr/cmip5/spip.php?article24**

CNRM-CM5 is one of the climate models that is part of the PMIP3. In our simulations it was one of climate forcings that lead to very small ice sheets. The Eurasian and North American ice sheets had no ice beyond the present-day ice coverage. Greenland partly melted when using CNRM-CM5 forcing which could be attributed to summer temperatures well above freezing. The temperature above Antarctica also shows strange patterns with warm regions on the ice sheet. Obviously, these results have a very large discrepancy with reconstructions. The website shown here shows a number of bugs found in CNRM-CM5. Point 19 states that the topography field (orog) is wrong. While we cannot find an issue in the CNRM-CM5 orog fields downloaded directly from the PMIP3 database, we cannot deny that there is discrepancy in LGM temperatures.

Despite these large discrepancies, we did decide on including the CNRM forced simulation in the manuscript. First of all, leaving out CNRM brings the question why we should not leave out MRI and GISS or any of the other GCM simulations. MRI has relatively high temperatures in Eurasia and North America, while GISS has low temperatures in most of Asia. Clearly also not a very good result. We believe, it is quite arbitrary to determine which GCM to omit fully from the manuscript. At the same time, it is also important to show the ice / paleo community that these long-timescale ice sheets simulations are very sensitive to climate forcings by the GCMs. There are large differences in the extent of the ice sheet that can be attributed directly to the climate forcing. By showing them all we hope to convey the message that the quality of the modelled ice sheet depends strongly on the quality of the climate forcing.

This is why we opted on using a subs-selection of GCM models for the main part of the analysis. After running the model using all nine PMIP3 GCMs, we found a large range of ice volumes. Some of these ice sheets had large discrepancies compared to reconstruction, so we made a selection of the climate models that were able to produce reasonable LGM ice sheets. Therefore, CNRM has in the end a minimal impact to the main analysis of the paper.